# Partitioning of the initial catalytic steps of leucyl-tRNA synthetase is driven by an active site peptide-plane flip

Luping Pang [1,2,3], Vladimir Zanki [4], Sergei V. Strelkov [1], Arthur Van Aerschot [2✉], Ita Gruic-Sovulj [4✉] & Stephen D. Weeks [1,5✉]

To correctly aminoacylate tRNA[Leu], leucyl-tRNA synthetase (LeuRS) catalyzes three reactions: activation of leucine by ATP to form leucyl-adenylate (Leu-AMP), transfer of this amino acid to tRNA[Leu] and post-transfer editing of any mischarged product. Although LeuRS has been well characterized biochemically, detailed structural information is currently only available for the latter two stages of catalysis. We have solved crystal structures for all enzymatic states of *Neisseria gonorrhoeae* LeuRS during Leu-AMP formation. These show a cycle of dramatic conformational changes, involving multiple domains, and correlate with an energetically unfavorable peptide-plane flip observed in the active site of the pre-transition state structure. Biochemical analyses, combined with mutant structural studies, reveal that this backbone distortion acts as a trigger, temporally compartmentalizing the first two catalytic steps. These results unveil the remarkable effect of this small structural alteration on the global dynamics and activity of the enzyme.

[1] Biocrystallography, Department of Pharmaceutical and Pharmacological Sciences, KU Leuven, Herestraat 49 – Box 822, 3000 Leuven, Belgium. [2] Medicinal Chemistry, Rega Institute for Medical Research, Department of Pharmaceutical and Pharmacological Sciences, KU Leuven, Herestraat 49 – Box 1041, 3000 Leuven, Belgium. [3] Research Center of Basic Medicine, Academy of Medical Sciences, College of Medicine, Zhengzhou University, Zhengzhou, Henan 450001, China. [4] Department of Chemistry, Faculty of Science, University of Zagreb, Horvatovac 102a, 10000 Zagreb, Croatia. [5] Pledge Therapeutics, Gaston Geenslaan 1, 3001 Leuven, Belgium. ✉email: arthur.vanaerschot@kuleuven.be; gruic@chem.pmf.hr; sweeks@pledge-tx.com

Adenylate-forming enzymes are a superfamily of structurally diverse proteins found in all three kingdoms of life which are involved in various essential cellular biological processes, including amino acid[1,2] and nucleoside/nucleotide biosynthesis[3,4], fatty acid metabolism[5–7], post-transcriptional modification[4] and aminoacylation[8,9] of tRNA necessary for protein biosynthesis. Based on the structural differences of the catalytic core, these enzymes can be clustered into a number of classes[10,11]. Despite being structurally and functionally distinct, they all generally catalyze a two-step reaction[10,11]. Carboxylate-containing substrates are first activated by condensation with adenosine-5′-triphosphate (ATP) to form a highly reactive and tightly bound acyl-adenylate intermediate (Acyl-AMP) with the concomitant release of inorganic pyrophosphate (PPi). In the second step, the Acyl-AMP is attacked by a nucleophile resulting in an amide, ester or thioester product and the liberation of AMP[10,11]. For the majority of adenylate-forming enzymes these distinct steps occur in a single active site[11,12]. Therefore, for efficient sequential catalysis, members of this superfamily must differentiate between the various substrates and coordinate their association based on intermediate product formation. How this is manifested at the atomic level is not known.

Mechanistically similar to other adenylate-forming enzymes, the class Ia aminoacyl-tRNA synthetase (aaRS) LeuRS charges tRNA^Leu isoacceptors, which are essential substrates for protein translation. LeuRS recognizes leucine (Leu) and ATP to form a leucyl-adenylate (Leu-AMP) intermediate, a reaction that proceeds via a pentavalent transition state[13–15]. Upon Leu-AMP formation, the acceptor arm of tRNA^Leu enters the synthetic site and the leucyl moiety is transferred to the 2′-OH group of the 3′-terminal ribose of this tRNA^Leu to generate the final product leucyl-tRNA^Leu (Leu-tRNA^Leu) with the release of AMP[8]. These steps are all catalyzed within the aminoacylation domain. This domain, which is commonly shared amongst other class I aaRSs, belongs to the HUP domain superfamily and topologically resembles a Rossmann fold (Fig. 1)[9,10,16,17]. In addition to charging the Leu-tRNA^Leu, LeuRS also has the capacity to hydrolyze incorrectly acylated tRNA^Leu species resulting from the mis-activation and subsequent aminoacyl transfer of non-proteinogenic amino acid[18,19]. This post-transfer editing occurs at a separate editing domain found inserted in a region of LeuRS dubbed the connective polypeptide 1 (CP1), which separates the aminoacylation domain into two halves at the primary sequence level[20–25] (Fig. 1). Crystal structures of bacterial LeuRS with tRNA^Leu bound in the aminoacylation and the editing conformation, have provided a detailed molecular view of the aminoacyl transfer and post-transfer editing stages[26,27]. In contrast, structural details corresponding to the first step of amino acid activation and their relationship to the conformation of the enzyme during aminoacyl transfer have remained elusive.

To date, our understanding of the reaction mechanism of aaRS, and the adenylate-forming superfamily as a whole, have in part come from pioneering studies characterizing the class Ic tyrosyl- and tryptophanyl-tRNA synthetases and the class Ib glutamyl- and glutaminyl-tRNA synthetases[14,28–35]. However, the atomic details of the adenylation reaction in the larger subclass Ia group, which also includes methionyl-, valyl-, isoleucyl- arginyl- and cysteinyl-tRNA synthetases have not been resolved[8,36]. Here we present high-resolution crystal structures of Neisseria gonorrhoeae LeuRS (NgLeuRS) in complex with various substrates representing the different enzymatic states during Leu-AMP formation. Comparison of these structures provides an atomic description of a dynamic cycle that involves the rearrangement of the active site and accessory domains. Intriguingly, a peptide-plane flip of a residue in an α-helix in the aminoacylation domain appears to play a key role in re-opening the active site following intermediate formation. Additional structural and enzymatic kinetic studies of NgLeuRS mutants show that this peptide-plane flip is necessary for the catalytically productive positioning of the 3′-end of the incoming tRNA^Leu as well as the temporal compartmentalization of the pre-transition and aminoacylation states. This work not only furthers our structural understanding of the whole catalytic cycle of LeuRS, but also illuminates one mechanism in the adenylate-forming enzyme superfamily for distinctly separating individual steps of catalysis within the same active site.

## Results

**NgLeuRS structures in different catalytic states**. To capture the different enzymatic states of the first catalytic step of LeuRS, crystals of the N. gonorrhoeae apoenzyme were soaked with Leu, ATP, ATP and L-leucinol, ATP and Leu. High-resolution diffraction data were collected for each complex and the structures were solved by molecular replacement using the structure of NgLeuRS complexed with an intermediate analog reported in our previous work (Fig. 2 and Supplementary Data 1)[37]. In each case unambiguous electron density was observed for the associated ligand (Supplementary Fig. 1a). Despite being derived from the same initial crystals it was only possible to trace the full protein sequence in the NgLeuRS·ATP·leucinol ternary complex (Fig. 1c). This structure shows the aminoacylation domain, which contains two signature sequence motifs (HIGH and KMSKS) that are highly conserved in all class I aaRSs[9,16], connected with six auxiliary domains: CP1 hairpin, editing, CP2 domain, leucine specific (LS) domain, anti-codon binding domain (ACBD) and the C-terminal domain (CTD) (Fig. 1b, c). The LS and CTD are typically disordered in the other solved structures, despite all having the same space group and a single molecule in the asymmetric unit (Supplementary Data 1).

As has been previously recognized, the active site of NgLeuRS is large (approximately 758 Å³ in the apo form), which is necessary for association of the various substrates. In all soaked structures substantial rearrangements of residues lining this site are observed (Fig. 2 and Supplementary Fig. 1b). Leu is present in a hydrophobic cavity surrounded by residues conserved across all species[17]. Compared to the apo structure, the binding of Leu induces the side chain of Y43, an invariant residue in all LeuRSs[26,38], to rotate 103.5° toward the Leu binding pocket to form a stacking interaction with the plane defined by the N-Cα-C of the Leu substrate (Fig. 2 and Supplementary Fig. 1b). The α-amino group of Leu forms H-bonds with the side chain Oδ2 of D80 and the main chain carbonyl oxygen of F41 (Fig. 2). In addition, a rearrangement of polypeptide loop region ^544GIEHA^548, which we will name the priming loop, is observed (Supplementary Fig. 1b, c). Compared to the apo state, the backbone of this region shifts a maximum of 2.8 Å bringing the side chains of the loop residues into the active site as a result of a H-bond between Nε1 of H547 and the carboxylate of Leu. This interaction is likely important for selecting the correct amino acid chirality.

The ATP-bound NgLeuRS structures were obtained by both soaking and co-crystallization (Fig. 2, Supplementary Fig. 1a and Supplementary Data 1). In both structures, the adenosine moiety can be fitted unambiguously and are almost identical. Binding of the nucleoside results in a 3.5 Å inward movement of the ^634KMSKS^638 loop (Supplementary Fig. 1b) because of a backbone interaction mediated by M635 with the N6 of the base. The position of the base moiety is additionally stabilized by H-bonds with the backbone atoms of V583 and a sulfur-π stacking interaction resulting from the inward rotation of the side chain of M582 (Fig. 2 and Supplementary Fig. 1b). Both the 2′ and 3′-OH of the ribose group form H-bonds with the backbone nitrogen of

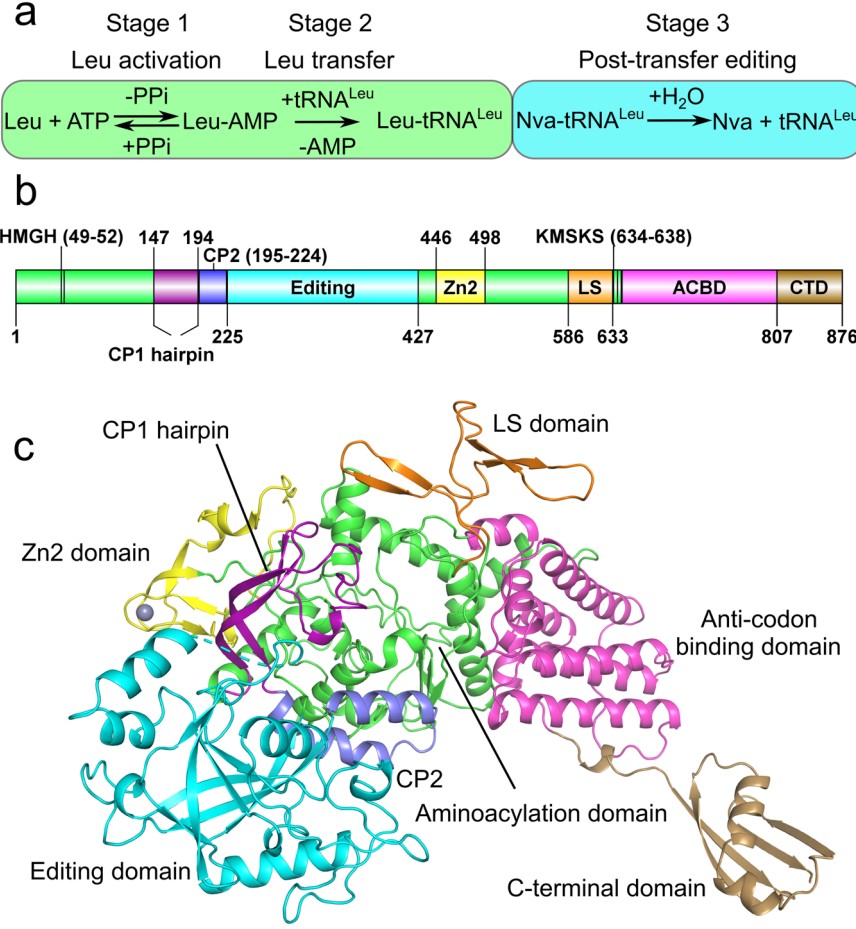

**Fig. 1 Biochemical and structural boundaries of *N. gonorrhoeae* LeuRS. a** The enzymatic steps catalyzed by LeuRS. The first two stages occur in the aminoacylation domain (green box) while the post-transfer editing of mischarged Nva-tRNA<sup>Leu</sup> is processed in the editing domain (cyan box). Nva is the abbreviation of non-canonical amino acid ʟ-norvaline. **b** The domain structure of *N. gonorrhoeae* LeuRS and **c** its corresponding cartoon representation. The aminoacylation domain (green) is split by multiple insertions: connective polypeptide 1 (CP1 hairpin, purple) and 2 (CP2, slate); the editing domain (cyan); the zinc-binding domain (Zn2, yellow) and the leucine specific domain (LS, orange). The aminoacylation domain is followed by the anti-codon binding domain (ACBD, magenta) and the C-terminal domain (CTD, brown). A zinc ion is shown as a gray sphere.

G544, which drives a similar shift of the priming loop [544]GIEHA[548] as seen for the Leu-bound state (Supplementary Fig. 1b). These two hydroxyl groups also interact with the side chain Nε2 of Q580 and the side chain Oδ1 of N55, respectively (Fig. 2). The former residue, which is highly conserved in LeuRS homologs across all kingdoms, specifically recognizes the ribose 2′-OH and is likely necessary for discriminating the ribose over the deoxyribose analog as a substrate. Although the overall protein backbone (all-atom RMSD of 0.137 Å) and the adenosine positions of ATP are nearly identical in the structures obtained by either soaking or co-crystallization, the triphosphate group of ATP in these structures is found in different positions. Surprisingly, neither of these two binding states of ATP are in productive conformation (Supplementary Fig. 1a). A Mg²⁺ ion has been proven to be important for octahedral coordination with the triphosphate group of ATP in class I aaRSs to promote the adenylation reaction[30,33,39,40]. Despite the presence of Mg²⁺ in the crystallization condition no density was observed for this ion.

**Large conformational changes accompany Leu-AMP formation.** To capture the pre-transition state, Leu and ATP were co-soaked with the NgLeuRS crystals, but only the intermediate Leu-AMP was observed in the active site indicating that the enzyme is active in this crystal form. Therefore, ʟ-leucinol, a reduced Leu

analog, and ATP were soaked together into equivalent crystals obtaining the NgLeuRS·ATP·leucinol structure. LeuRS, together with IleRS, ValRS, and MetRS is a class Ia aaRS[8,9,36,41], thus to the best of our knowledge this ternary structure provides the first structural insights into the pre-transition state of this aaRS subclass. In this complex, leucinol and the adenosine moiety of ATP are recognized in the same manner as seen for Leu or ATP alone (Fig. 2). The triphosphate group of ATP is fully ordered, coordinating a Mg²⁺ ion and making extensive interactions with residues of the class I signature HIGH motif which has the sequence of [49]HMGH[52] in NgLeuRS, with the KMSKS loop and the CP1 hairpin (Fig. 2). This results in a more dramatic movement of KMSKS loop into the aminoacylation site with a maximum distance of 8.6 Å compared to apo-form structure (Supplementary Fig. 1b). The α-phosphate is H-bonded with the side chain Nε2 of H52 and Nζ of K637 and backbone nitrogen of Y43. The β-phosphate is stabilized by forming H-bonds with the side chain Nε2 of H49, Nζ of K634 and K637, and the backbone of K637, while the γ-phosphate establishes a salt bridge with R178 from the CP1 hairpin and forms H-bonds with side chain Oγ and main chain nitrogen of S638. In addition, the α, β, γ-phosphates of ATP and three additional ordered water molecules coordinate a Mg²⁺ ion, forming an ideal octahedral geometry with an average distance around 2 Å between Mg²⁺ and the contacting atoms (Fig. 2 and Supplementary Fig. 1d).

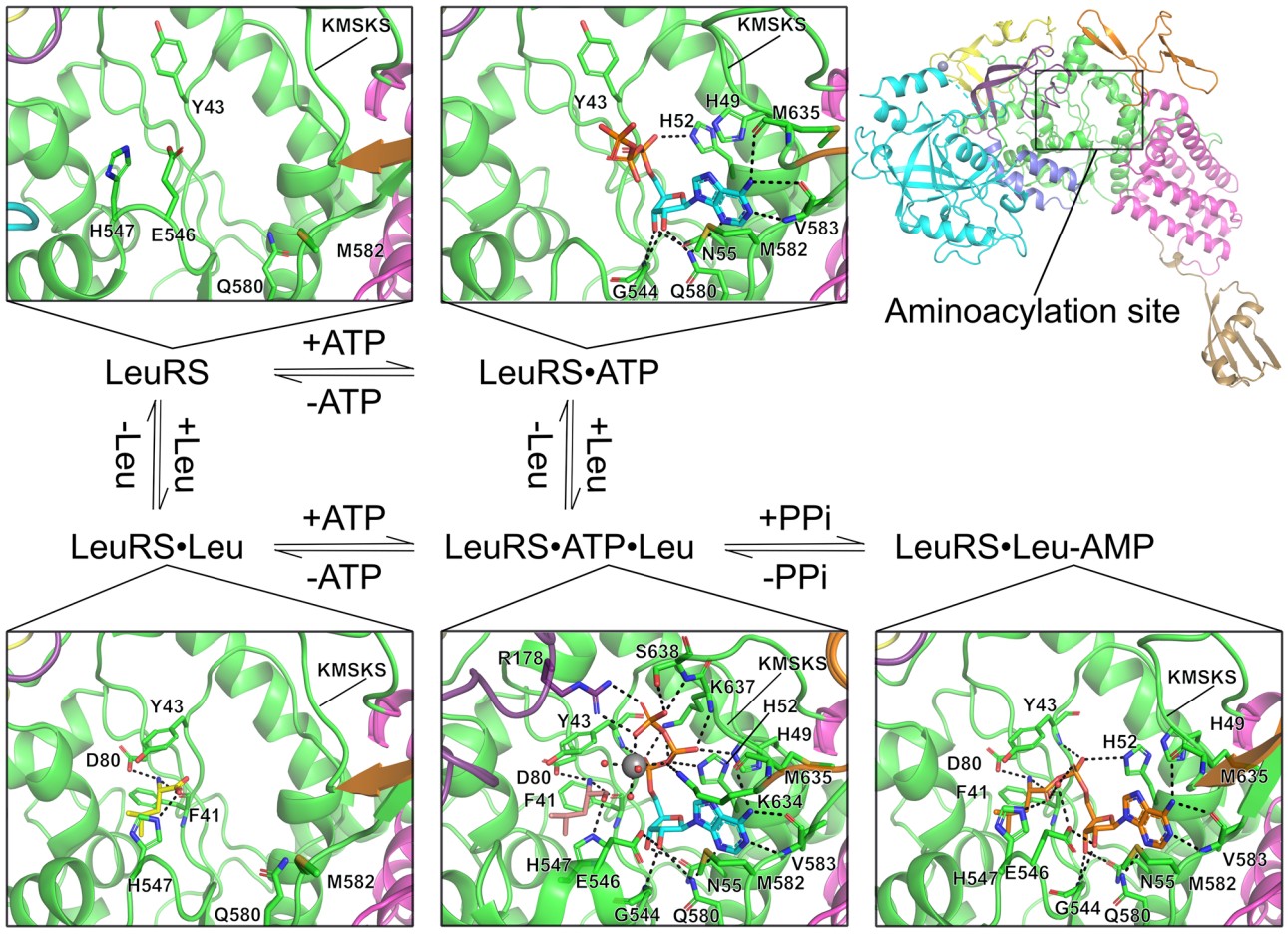

**Fig. 2 Snapshots of each catalytic step during Leu-AMP formation.** All protein structures are shown as cartoon representations whereas ligands and interacting protein residues are shown as sticks. All H-bonds and salt bridges are shown as black dashed lines. A magnesium ion is shown as a gray sphere while its coordinating water molecules are shown as small red spheres. All structural domains are colored as shown in Fig. 1c. Substrates ATP (cyan), Leu (yellow), Leu analog L-leucinol (salmon), and the reaction intermediate Leu-AMP (orange) are shown as sticks. The NgLeuRS·ATP·leucinol ternary complex mimics the pre-transition state where both ATP and Leu are bound in the productive conformation in the aminoacylation site.

Superposition of the Leu-bound structure onto the pre-transition state model shows that the carboxylate oxygen atom which is in the *syn* conformation with the α-amino group of the Leu substrate is located in the required position to allow nucleophilic attack on the α-phosphate of ATP at a (O-P) distance of 2.9 Å (Supplementary Fig. 1d). This positioning is in good agreement with the theoretical computational models that show the *syn* oxygen atom is the preferred attacking atom in class I aaRSs[42]. The other carboxylic oxygen of the substrate Leu, equivalent to the hydroxyl group of leucinol, and α-phosphate oxygen directly interacts with one of three structural water molecules further assisting in the stabilization of the water shell surrounding $Mg^{2+}$ and suggesting the synergistic binding of ATP and Leu, in the presence of $Mg^{2+}$. In addition, superposition of *Geobacillus stearothermophilus* TrpRS·ATP·tryptophanamide and our NgLeuRS ternary structure based on the adenosine moiety of ATP (RMSD 0.09 Å for all ATP atoms) shows that the triphosphate group of ATP in both structures shares a very similar extended conformation with the triphosphate of ATP equivalently coordinating with a $Mg^{2+}$ ion. However, the three structured water molecules surrounding $Mg^{2+}$ seen in the NgLeuRS structure are not observed in the former complex, suggesting that the NgLeuRS·ATP·leucinol structure can be a better model of the pre-transition state of other class I aaRSs (Supplementary Fig. 1e).

In comparison to the above structures, in the Leu-AMP bound complex the leucyl and AMP moieties make the same interactions

as the single substrates alone. Relative to the pre-transition state, the release of PPi byproduct leads to the loss of the corresponding interactions with KMSKS loop and R178 of the CP1 hairpin. Consequently, the KMSKS loop is located in the same semi-open conformation as observed in the ATP-bound structure (Fig. 2 and Supplementary Fig. 1b).

Zooming out and focusing on the global changes, the transition between the different enzyme-bound states results in considerable domain rearrangements. Comparison of the NgLeuRS apo structure to that of the NgLeuRS·ATP·leucinol ternary complex shows that the CP1 hairpin (residues 147–194) together with a structurally associated α-helix (residues 84–93) rotates 31° (Supplementary Fig. 2a) toward the active site pushing the editing domain (residues 225–427) 29° away from the same region. The LS domain along with KMSKS loop (residues 586–638) are simultaneously rotated 38° inward forming a more compact active site (Fig. 3a). This feature brings in conserved interactions with the triphosphate moiety of ATP and allows the carboxylate group of Leu to be positioned close enough for the subsequent reaction with the α-phosphate of ATP (Supplementary Fig. 1d).

**A peptide-plane flip drives global conformational changes.** Intriguingly, careful examination of the different X-ray structures shows the presence of a peptide-plane flip between L550 and

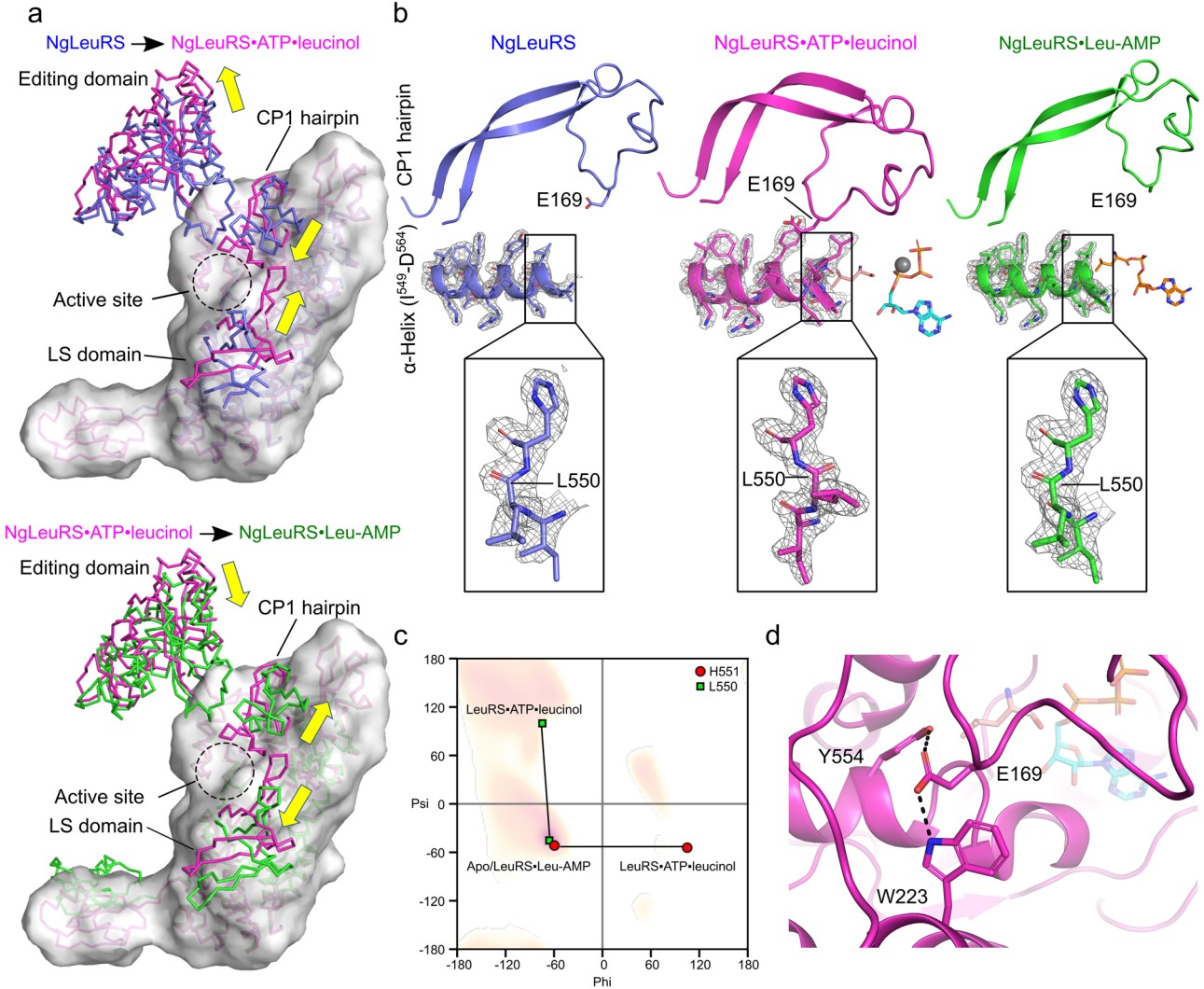

**Fig. 3 Structural comparison of different catalytic states of NgLeuRS. a** Top, ribbon representation of the superposition of the ternary structure of NgLeuRS·ATP·leucinol (magenta) with NgLeuRS (slate). Bottom, superposition of the ternary structure of NgLeuRS·ATP·leucinol (magenta) with NgLeuRS·Leu-AMP (green). In both panels the aminoacylation domain, CP2, Zn2, ACBD and CTD of the NgLeuRS·ATP·leucinol structure are additionally shown as a surface representation. **b** Top panels, the relative position of the CP1 hairpin (cartoon backbone representation) to the L550-containing α-helix (cartoon and stick representation) in different catalytic states. Ligands are shown as sticks representations while the $Mg^{2+}$ ion is shown as a gray sphere. Bottom, zoom of the L550 region. 2Fo-Fc electron density maps countered at 1.5 σ (gray mesh). **c** Comparison of the Ramachandran plot of L550 (green square) and H551 (red circle) in NgLeuRS/NgLeuRS·Leu-AMP and NgLeuRS·ATP·leucinol structures. **d** Polar interactions between E169 in the CP1 hairpin and surrounding protein residues in the NgLeuRS·ATP·leucinol complex. H-bonds are shown as black dashed lines.

H551 in the NgLeuRS·ATP·leucinol complex, that is absent in the apo and the intermediate bound structures (Fig. 3b). These residues are found at the N-terminus of a structurally conserved α-helix (residues 549–564) that lines the substrate binding pocket, and the peptide-plane flip disrupts the secondary structure of these helix capping residues. This transition is clearly reflected in the Ramachandran plots, which show that L550 moves from a right-handed α-helix (αR) to β-sheet position, while H551 moves from αR region to an unfavorable zone upon formation of the pre-transition complex (Fig. 3c).

The NgLeuRS·ATP·leucinol structure suggests that the peptide-plane flip is induced by the closure of the CP1 hairpin in order to avoid the steric clash between the side chains of L550 and the CP1 hairpin E169. Compared to the individual substrate-bound states, the energetically unfavorable peptide-plane flip and closed conformation of the CP1 hairpin in the pre-transition state is stabilized by a salt bridge between R178 from the CP1 hairpin and the γ-phosphate of ATP and polar interactions between the side

chain of E169 from the CP1 hairpin and the side chains of Y554 from L550-containing helix and W223 located at the C-terminus of the CP2 domain (Fig. 3d). In addition, this peptide-plane flip is further accommodated by the movement of the priming loop 544GIEHA548 induced by substrate binding (Supplementary Fig. 1b).

To understand the role of this peptide-plane flip on the global conformational changes and its effects on enzymatic catalysis, L550 was mutated to an alanine or glycine and the crystallization of both mutants was performed. These mutants gave rise to the same crystal form as the WT protein by seeding. However, while the NgLeuRS-L550A diffracted as well as WT protein crystals, the NgLeuRS-L550G showed slightly lower diffraction quality in the range of 2.7–3.3 Å (Supplementary Data 1). Similar to WT, soaking of the crystals with ATP and Leu resulted in the formation of Leu-AMP indicating that both mutants retain the capability to carry out the first step of the aminoacylation reaction (Fig. 4, Supplementary Data 1 and Supplementary Fig. 3a, b).

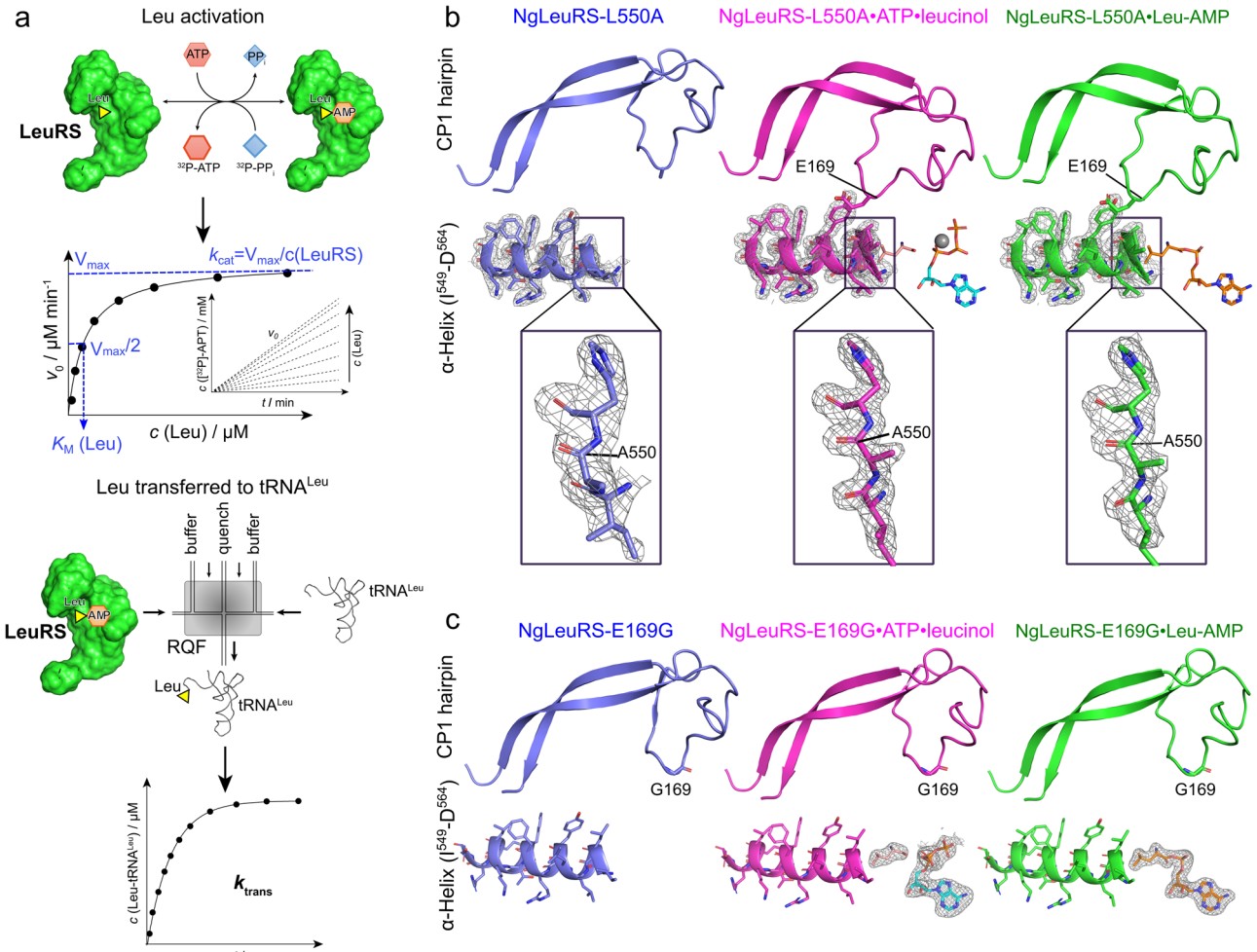

**Fig. 4 The effects of L550A and E169G mutants on NgLeuRS catalytic efficiency. a** Schematics of assays for Leu activation (top) and Leu transfer (bottom). The former is performed by ATP-PPi exchange assay to measure the effects of L550A and E169G mutants on Leu activation while the latter was used to evaluate the second stage Leu transfer to tRNA$^{Leu}$. **b** Upper panels: the relative position of the CP1 hairpin to the L550A-containing α-helix in the NgLeuRS-L550A mutant in different trapped catalytically states. Lower panels: zoom of the L550A region. The CP1 hairpin remains in the closed conformation following Leu-AMP formation. 2Fo-Fc electron density maps (gray mesh) are all countered at 1.5 σ. **c** Relative position of CP1 hairpin to the L550-containing α-helix in NgLeuRS-E169G mutant. The CP1 hairpin of the E169G mutant remains in an open conformation during Leu activation. Omit maps (gray mesh) of ligands are countered at 3.5–5 σ. In **b**, **c** the CP1 hairpin is shown as cartoon backbone representations while the L550-containing α-helix is a mixed cartoon and stick representation. The ligands are shown as sticks while the catalytic magnesium ion is shown as a gray sphere.

Comparison of the captured apo and pre-transition states of both mutants show the same large domain movement, including closure of the active site by the CP1 hairpin and structurally associated α-helix (M$^{84}$-N$^{93}$), that were seen for the WT structures. Despite the occurrence of these substrate-induced changes, including the shift of the priming loop, no peptide-plane flip was observed in either mutant pre-transition state structure. This is likely the result of the absence of the clash between the side chains of E169 of the CP1 hairpin and A550/G550 although the side chain of E169 maintains the same interactions as seen for the WT pre-transition structure (Fig. 4 and Supplementary Fig. 3a, b).

While the apo and pre-transition states globally look similar to the equivalent structures of the WT protein, for Leu-AMP bound structures of both NgLeuRS L550 mutants, the CP1 hairpin and associated α-helix (M$^{84}$-N$^{93}$) are found in the closed conformation (Supplementary Fig. 3a, b). These structures are superposable to the corresponding pre-transition state of both mutants, whereas the CP1 hairpin in the equivalent Leu-AMP bound structure for WT is opened and resembles the apo state (Figs. 3a and 4b and Supplementary Fig. 3a, b). This suggests that when

soaking ATP and Leu in WT crystals, the CP1 hairpin closes during the Leu activation reaction but then repositions once this reaction is completed. The failure of the CP1 hairpin to reopen in both mutants appears to be correlated with the absence of the peptide-plane flip. When compared to available structures of EcLeuRS in complex with tRNA$^{Leu}$ mimicking the aminoacylation state[26] (PDB ID: 4AQ7), this closed state is incompatible with the entry of the acceptor arm of the cognate tRNA$^{Leu}$ for the second step of aminoacylation and therefore may have an effect on the full aminoacylation reaction.

To further investigate the catalytic role of the L550 peptide-plane flip, the NgLeuRS-L550G mutant was tested for Leu-tRNA$^{Leu}$ formation under steady-state (multi-turnover) conditions. A 25-fold decrease in the aminoacylation rate constant was observed (5.8 s$^{-1}$ vs 0.26 s$^{-1}$; Table 1) indicating an important role of the peptide-plane flip in LeuRS turnover. However, both catalytic steps: Leu activation and leucyl transfer to tRNA$^{Leu}$, may influence the rate of aminoacylation turnover. To localize the catalytic role of the peptide-plane flip, we independently tested amino acid activation by an ATP-PPi exchange assay, and the aminoacyl transfer step by single-turnover experiments (Fig. 4a and Supplementary Fig. 4). In the first

**Table 1 Kinetic parameters for wild type and mutant LeuRSs from *N. gonorrhoeae* and *E. coli*.**

| Enzyme | | Amino acid activation | | | | | Aminoacyl transfer | Aminoacylation |
|---|---|---|---|---|---|---|---|---|
| | | Leu | | | ATP | | | |
| | $k_{cat}$ (s$^{-1}$) | $K_M$ (μM) | $k_{cat}/K_M$ (s$^{-1}$ μM$^{-1}$) | | $K_M$ (mM) | $k_{cat}/K_M$ (s$^{-1}$ mM$^{-1}$) | $k_{trans}$ (s$^{-1}$) | $k_{obs}$ (s$^{-1}$) |
| EcLeuRS WT | 66.7 ± 0.8 | 31.4 ± 2.9 | 2.12 | | 3.3 ± 0.3 | 20.21 | 58 ± 5[a] | 4.9 ± 0.2 |
| EcLeuRS-M536G | 34.2 ± 1.7 | 52.3 ± 5.9 | 0.65 | | 7.4 ± 0.8 | 4.62 | 15.0 ± 0.8 | 1.41 ± 0.03 |
| NgLeuRS-WT | 90 ± 1 | 14.3 ± 0.8 | 6.3 | | 2.5 ± 0.2 | 36.0 | 52 ± 4 | 5.8 ± 0.4 |
| NgLeuRS-L550G | 54.2 ± 1.3 | 15.3 ± 1.0 | 3.54 | | 2.9 ± 0.3 | 18.69 | 1.7 ± 0.1 | 0.26 ± 0.02 |
| NgLeuRS-E169G | 21.7 ± 1.3 | 76.6 ± 7.4 | 0.28 | | 5.8 ± 0.8 | 3.74 | 78 ± 7 | 1.35 ± 0.08 |

The values represent the mean ± SEM of at least three independent experiments.
[a]Obtained by EcLeuRS with inactivated editing domain[43]. The transfer rate using EcLeuRS WT determined in this work was 65 s$^{-1}$.

assay NgLeuRS-L550G showed only a twofold decrease in the $k_{cat}/K_M$ for Leu activation relative to the WT protein (Table 1). Thus, even though the peptide-plane flip is not observed in the pre-transition structure of this mutant, the kinetic data clearly show that the absence of this backbone alteration does not impose a kinetic penalty during activation. Next, we isolated and separately tested the leucyl transfer step. The transient kinetic analysis unambiguously demonstrated a kinetic defect manifested as a 31-fold drop in the single-turnover rate constants for L550G mutant as compared with WT NgLeuRS (1.7 s$^{-1}$ vs 52 s$^{-1}$, Table 1). Therefore, our data unveil the peptide-plane flip as an essential part of the second step of the aminoacylation reaction, with a proposed role in re-opening of the synthetic site that accompanies Leu-AMP formation and is a prerequisite for productive positioning of the 3′-end of tRNA$^{Leu}$.

Multi-genome analysis of bacterial LeuRSs shows high sequence and predicted secondary structure similarity, both within the L550-containing α-helix, and the surrounding active site residues (Supplementary Fig. 5). Interestingly, L550 is not fully conserved among all LeuRSs and is substituted by a methionine residue in some enzymes (e.g. *Escherichia coli* LeuRS). EcLeuRS has been well characterized[18,19,43] and the overall full-length sequence identity and similarity between NgLeuRS and EcLeuRS are 57 and 71%, respectively[10]. To see if M536 in EcLeuRS has the same role as L550 in NgLeuRS, EcLeuRS-M536G was produced and kinetically tested as described above. The effect of M536G mutation in EcLeuRS parallels the effect of L550G mutation in NgLeuRS showing threefold defect in $k_{cat}/K_M$ during the activation step that is accompanied by a fourfold decrease in the rate of the leucyl transfer step (Table 1 and Supplementary Fig. 4). Overall (two-step) aminoacylation is decreased by threefold (Table 1). Taken together, the M536G mutation in EcLeuRS produced similar but less disruptive effects as compared with L550G in NgLeuRS, suggesting that the use of the peptide-plane flip for catalysis could be shared among other bacterial LeuRSs (Table 1). These combined results demonstrate that the peptide-plane flip of L550 drives the re-opening of the CP1 hairpin upon Leu-AMP formation which is essential for the subsequent entry of the 3′-end of tRNA$^{Leu}$. Therefore, this flip appears to act as a catalytic-step-dependent trigger that promotes the acyl transfer reaction.

**The CP1 hairpin swing is mediated by L550-E169 partnership.** As the substrate-induced peptide-plane flip results from the clash with the incoming side chain of E169 (Fig. 3b), we next generated a NgLeuRS-E169G mutant to investigate its corresponding effects on aminoacylation. The mutant was readily expressed and crystallized, and high-resolution structures of the NgLeuRS-E169G apo, and soaked NgLeuRS-E169G·ATP·leucinol and NgLeuRS-E169G·Leu-AMP were determined (Supplementary Data 1). These three structures can be fully overlaid with each other (and

WT apo) with an all-atom RMSD of 0.186 Å (Supplementary Fig. 3c) indicating no obvious conformational changes between apo, pre-transition and intermediate states which strongly contrasts with what was observed for the WT enzyme. This is reflected in the fact that in all states the CP1 hairpin remains in the open conformation (Fig. 4c). In the case of NgLeuRS-E169G·ATP·leucinol complex this results in the loss of the essential salt bridge between the side chain of R178 and γ-phosphate group of substrate ATP (Figs. 2 and 4c). Consequently, the triphosphate group of ATP is in a flexible non-productive conformation as implied by its high crystallographic B-factors compared to the equivalent WT complex. In particular, the carboxylate oxygen atom of Leu is incapable of reacting with the α-phosphate of ATP residing at a distance of 3.7 Å. However, when the crystal is soaked with Leu and ATP, the intermediate is still formed as evidenced by the electron density map (Fig. 4c). This suggests that despite the high mobility of triphosphate group of ATP, the mutant enzyme is still able to reach the productive conformation and complete the reaction within the two hours of soaking of the crystals with ATP and Leu at concentrations that exceed their respective $K_m$ values.

The inability of the bound L-leucinol and ATP to induce closure of the CP1 hairpin in the E169G mutant, despite the removal of the L550 clashing side chain, suggests that the substrates alone cannot promote the closed active site conformation. The interactions mediated by E169 with Y554 and W223 (Fig. 3d) are thus additionally required to stabilize the closed CP1 hairpin position in the pre-transition state to promote the formation of Leu-AMP. In line with this model, independently tested amino acid activation demonstrated that substitution of E169 with a glycine results in up to fivefold increase in $K_M$ (Leu) and fourfold decrease in $k_{cat}$ yielding the 24-fold less efficient enzyme in Leu activation relative to the WT protein (Table 1 and Supplementary Fig. 4). Thus, opposite to the L550G mutant which captures the "closed conformation" and displays a defect in the aminoacyl transfer step, mutation at the CP1 hairpin which stabilizes the "open conformation" affects only the activation step.

## Discussion
In this work, we have revealed five NgLeuRS structures representing each enzymatic state during Leu activation. In addition to these, crystal structures mimicking the aminoacylation state with the 3′-end of tRNA$^{Leu}$ in the synthetic site have been reported for EcLeuRS[26] (PDB ID: 4AQ7) and *Pyrococcus horikoshii* LeuRS[44] (PDB ID: 1WZ2), and structures representing the post-transfer editing conformation with the 3′-end of tRNA$^{Leu}$ in the editing site have been described for *E. coli* and *Thermus thermophilus* LeuRSs[26,27]. Since *N. gonorrhoeae* LeuRS shows a high sequence and structural similarity with *E. coli* and *T. thermophilus* ortho-logues, the tRNA$^{Leu}$ bound LeuRS structures from the latter two

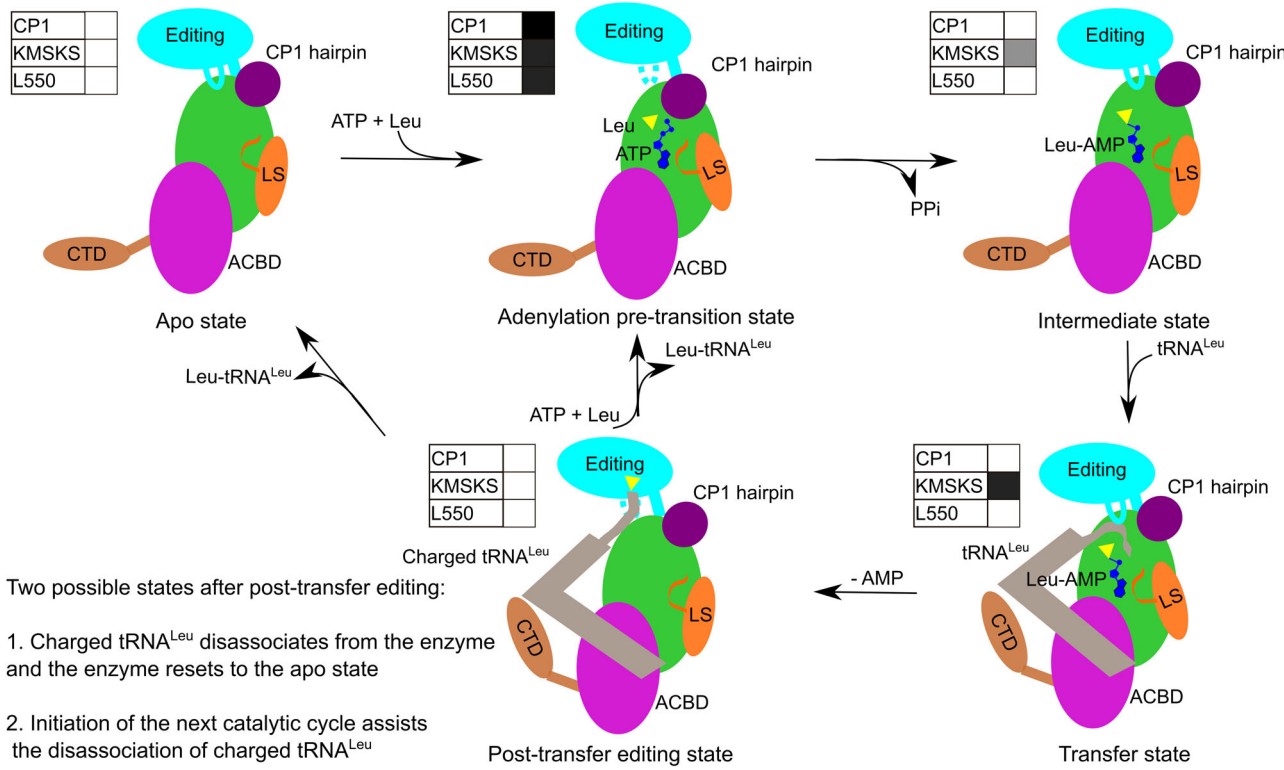

**Fig. 5 The proposed structural dynamics during the catalytic cycle of NgLeuRS.** Color coding for each structural domain is shown the same as Fig. 1c with Leu represented as a yellow triangle, the adenosine and phosphate groups are dark blue and tRNA[Leu] is colored in gray. The C-terminal domain is highly dynamic and poorly ordered before tRNA[Leu] binding. The conformational state of the CP1 hairpin, the KMSKS loop and associated LS domain, and the peptide plane of L550 are shown in the left corner boxes for each catalytic step. White represents an open conformation or a regular peptide plane, while the black box represents a closed state or flipped peptide plane, and gray represents a semi-open conformation.

bacterial species provide complementary mechanistic information allowing us to propose a full structural description of the whole catalytic cycle of NgLeuRS (Fig. 5).

In the apo state, the synthetic site of NgLeuRS is fully open allowing the entry of all substrates. In the amino acid activation step, the association of both Leu and ATP results in a dramatic compaction of this site by inducing protein rearrangements at three distinct faces of the pre-transition state structure. These movements involve the CP1 hairpin and associated α-helix ($M^{84}$-$N^{93}$) that are in proximity of the Y43 capped Leu binding site, the conserved KMSKS loop and LS domain close to the adenine binding site, and the priming loop that borders the ribose (Fig. 5 and Supplementary Fig. 2a). Combined, these changes result in a greater than 60% reduction of the active site volume that protects and stabilizes the pre-transition state and excludes the entry of the acceptor stem of tRNA[Leu]. In this high complementarity state, the carboxylate oxygen atom of Leu is positioned for nucleophilic attack on the α-phosphate of ATP. Crucially, the initial substrate-induced macroscopic rearrangements lead to a peptide-plane flip of L550, caused by the steric occlusion of the side chain of this residue by that of E169, the latter being present on the zinc knuckle of the incoming CP1 hairpin. Our biochemical data show that in the absence of steric constraints the peptide-plane flip is not necessary for the amino acid activation. Instead, the flip back to its favored position once the intermediate is formed, enables the aminoacyl transfer step by promoting the repositioning of the CP1 hairpin. Additionally, formation of the intermediate results in a relaxation of the KMSKS loop leading to a semi-open active site (Supplementary Fig. 1b). The combined re-opening of the active site permits the release of the PPi byproduct and allows the entry of the acceptor arm of the cognate tRNA[Leu].

During Leu activation, our crystal structures show an anti-parallel movement of the editing domain relative to that of the CP1 hairpin. The former has to move out of the way to accommodate the substrate-induced closure of the latter. Based on the structure of EcLeuRS in complex with tRNA[Leu] in the aminoacylation state[26] (PDB ID: 4AQ7), we propose that after the intermediate is formed, the CP1 hairpin is repositioned as a result of the L550 peptide-plane flip and the editing domain can move toward the active site bringing in the 3′-end of tRNA[Leu] (Fig. 5). The side chain of Y43, which is stacked on the leucyl moiety of Leu-AMP, switches to an open conformation that provides the space for the binding and reaction with the 3′-terminal adenosine of tRNA[Leu]. The KMSKS loop and connected LS domain fully close again to stabilize the tRNA[Leu] positioning. In addition, a loop region of the editing domain (residues $G^{297}$-$M^{307}$ in NgLeuRS) stacks against the CCA tail of tRNA[Leu] acceptor arm. A conserved glutamate (E292 in EcLeuRS corresponding to E301 in NgLeuRS) in this loop region forms an essential salt bridge with an invariant arginine (R416 in EcLeuRS and R428 in NgLeuRS) in the C-terminus of the editing domain that further stabilizes the 3′-end of tRNA[Leu] in the synthetic site[26,45]. Once this complex is formed, the nucleophilic reaction is proposed to proceed by assistance of a $Mg^{2+}$ co-factor, the α-amino group of Leu and two conserved amino acid residues (D80 and E532 in EcLeuRS corresponding to D80 and E546 in NgLeuRS)[26,46]. In this step, the essential salt bridge described above constrains the 3′-end of tRNA[Leu] in the synthetic site and partitions the aminoacylation and post-transfer editing processes.

To ensure fidelity following Leu-tRNA[Leu] formation, the charged 3′-end of tRNA[Leu] is translocated to the editing domain, where mischarged products are hydrolyzed[21,47,48]. Previously, it

was proposed that positioning of the tRNA$^{Leu}$ in the editing conformation is promoted by the reclosure of the CP1 hairpin and movement of KMSKS associated LS domain[26,27]. However, our structures of the initial step of aminoacylation show that the movement of the CP1 hairpin is highly dependent on the binding of Leu and ATP (Fig. 3 and Supplementary Fig. 2a). The presented structures rather point to a model where the translocation of the charged tRNA$^{Leu}$ to the editing site could be the result of movements of the priming and KMSKS loops (Fig. 5). Upon leucyl transfer to tRNA$^{Leu}$, the loss of synthetic site interactions with the aminoacyl moiety and the release of AMP lead to the return of the priming loop to its initial apo conformation (Fig. 5). Superposition of our NgLeuRS apo structure with that of the homologous *E. coli* enzyme bound to tRNA$^{Leu}$ (PDB ID: 4AQ7) suggests that the repositioning of the priming loop would lead to a steric clash with the 3′-end of tRNA$^{Leu}$ forcing the acceptor arm of this large substrate out of the synthetic site (Supplementary Fig. 2b). Furthermore, a loss of substrate interactions with the KMSKS loop would result in repositioning of this region and the structurally associated tRNA$^{Leu}$ binding LS domain. Together these changes would result in expulsion of the acceptor arm of the tRNA$^{Leu}$ from the synthetic site allowing it to move toward the editing site of the editing domain.

Once formed, the release of the charged tRNA product has been shown to limit the overall rate of two-step aminoacylation in class I aaRSs[49]. This observation is further supported by the kinetic analysis data of NgLeuRS in this work (Table 1). Interestingly, the NgLeuRS mutants mimic the WT enzymes in having rate-limiting product release (Table 1). Accordingly, the observed drop in mutants' aminoacylation rates (for example 1.35 s$^{-1}$ vs 5.8 s$^{-1}$ for E169G and WT) additionally suggests that the introduced mutations also slow down the product release step. This in turn indicates that the mobility of CP1 hairpin influenced by both L550 and E169 appears to be important for the disassociation of the final product. Detailed studies of IleRS, an enzyme belonging to the same aaRS subclass as LeuRS have demonstrated that the rebinding of Ile and ATP, or formation of the catalytic cycle intermediate Ile-AMP, occurs first and promotes the release of Ile-tRNA$^{Ile}$ [50–52]. Similarly, we hypothesize that LeuRS may also start the next cycle of Leu activation leading to closure of the CP1 hairpin which promotes the dissociation of Leu-tRNA$^{Leu}$ (Fig. 5).

Taken together, the repositioning of the substrate-induced L550 peptide-plane flip drives the re-opening of CP1 hairpin and promotes the subsequent aminoacylation reaction. This peptide-plane flip therefore functions as a temporal marker, triggering the switch between an active site conformation necessary for Leu activation, to that required for the subsequent aminoacyl transfer step. Additionally, the triggered movement of the CP1 hairpin not only plays an essential role in aminoacylation but also is important in the final product release. Thus, this subtle backbone alteration, and the correlated large-scale domain motions it induces, are important in modulating the multi-step NgLeuRS catalysis. As the associated components are conserved amongst bacterial LeuRS homologs the critical role of this peptide-plane flip is likely shared (Supplementary Fig. 5 and Table 1).

A systematic examination of peptide-plane flips in available crystal structures has identified that most are located in mobile loop regions, and typically occur for glycine residues[53]. In contrast, the peptide-plane flip of L550 observed in the pre-transition state structure of NgLeuRS is quite exceptional, being highly energetically unfavorable because of the distortion of the structurally conserved secondary structure of the α-helix where this residue resides (Fig. 3). Peptide-plane flips are known to have an important role during protein folding but only an extremely limited number of examples have been observed to modulate enzymatic activity. A peptide-plane flip in ubiquitin allosterically

affects the switch of expansion and contraction of the entire protein which in turn modulates its binding to a ubiquitin-specific protease, although not being directly involved in the interface with the latter[54]. Another example can be found in flavodoxins, where a peptide-plane flip has been observed between oxidized and reduced states and is important for stabilization of the semiquinone and modulating redox potential[55]. In both examples the associated peptide-plane flip involves a residue pair where one of the amino acids is a glycine. The use of this more flexible residue is in stark contrast to the large side chains of L550-H551 in NgLeuRS, that must accommodate a transition to the unfavorable region of the Ramachandran plot. This transition though is crucial in defining the properties of this active site switch.

In summary, our structural characterization of *N. gonorrhoeae* LeuRS provides a high-resolution structural framework providing insight into the amino acid activation mechanism of class Ia aaRSs. This study demonstrates that the activation of Leu with ATP by LeuRS involves a cycle of dramatic conformational changes, involving multiple domains, that parallels motions seen during post-transfer editing[26]. The structures, supported by biochemical studies, point to the unique role of a peptide-plane flip in acting as a switch that partitions the first two catalytic steps of this enzyme. To the best of our knowledge, this is the first example of a partitioning mechanism for a member of the adenylate-forming enzyme superfamily. It highlights how, at the molecular level, an enzyme uses the same synthetic site for distinct catalytic stages, while transitioning between these steps in a controlled manner.

## Methods

**Preparation of *N. gonorrhoeae* and *E. coli* LeuRSs.** The encoding sequence of full-length *N. gonorrhoeae* LeuRS was amplified by a one-step PCR from isolated genomic DNA. The reaction was conducted by using the high-fidelity Q5 DNA polymerase (New England Biolabs, Ipswich, MA, USA) with the forward primer 5′-gcgaacagattggtggtggt**ggt**ATGCAAGAACATTACCAGCCCG-3′ and the reverse primer 5′-ttgttagcagaagcttaTTAGACGACGATGTTCACCAGT-3′. The small-caps bases represent the adapter sequence complementary to plasmid cloning site, while the large-cap bases are the annealing sequences and the underscored triplet correspond to a non-native glycine residue, which was inserted before the start codon of NgLeuRS to facilitate the cleavage of the SUMO tag during purification. The PCR product was visualized and separated by an agarose gel, followed by gel extraction and in-fusion ligation with the linearized pETRUK vector containing an N-terminal pI enhanced SUMO tag as described in previous publication[37]. The recombinant plasmid pETRUK-NgLeuRS was first propagated in *E. coli* NEB5α and then verified through DNA sequencing (LGC genomics, Berlin, Germany). Three mutant constructs of pETRUK-NgLeuRS-L550A, pETRUK-NgLeuRS-L550G and pETRUK-NgLeuRS-E169G were generated by site-directed mutagenesis. Briefly, forward and reverse primers were designed with around 16 base pairs (bp) overlap containing the mutation. These primers were used for PCR-based amplification using the pETRUK-NgLeuRS plasmid as a template. Following DpnI treatment to digest the template, the PCR product was further separated by DNA electrophoresis. Purified PCR products were then directly transformed into *E. coli* NEB5α. The mutant genes were sequenced to confirm they contained only the desired single-site mutations.

The NgLeuRS and its corresponding mutants were expressed and purified using an identical protocol as described in our prior study[56]. The NgLeuRS construct was transformed into *E. coli* Rosetta 2 (DE3) pLysS expression host. A single colony was inoculated into 2 mL LB medium supplemented with 100 μg/mL ampicillin and 30 μg/mL chloramphenicol and grown at 37 °C with 250 rpm for 8 h. Then the preculture was inoculated into 2 L ZYP-5052 auto-induction medium[57] with addition of a final concentration of 1 mM ZnSO$_4$ (Sigma Aldrich, Burlington, USA) and 1 mL of antifoam SE-15 (Sigma Aldrich, Burlington, USA). The culture was grown overnight at 24 °C and when an OD$_{600nm}$ of 4.0 was reached, the temperature was decreased to 18 °C. After another 24 h of growth, cells were harvested by centrifugation at 7500 × *g* and 4 °C and the resulting cell pellets were resuspend in cation exchange buffer A (CEXA) containing 25 mM HEPES-NaOH pH 8, 200 mM NaCl, 5 mM β-mercaptoethanol (β-ME) and frozen at −80 °C.

For protein purification, cell lysis was initiated by thawing the cell pellet along with further dilution with CEXA in a final 8:1 v/w (CEXA: weight of pellet) ratio and the addition of 100 U cryonase cold-active nuclease (Takara, Shiga, Japan) and 10 mM MgCl$_2$. Cells were lysed by sonication on ice and then clarified by centrifugation at 18,000 × *g* at 4 °C for 30 min. The clarified supernatant was loaded onto a 5 mL Hitrap SP HP column (Cytiva, Marlborough, MA, USA). SUMO tag

fused NgLeuRS protein bound onto the column and was eluted by applying a linear gradient 0–50% cation exchange buffer B (CEXB) comprising of 25 mM HEPES-NaOH pH 8, 1000 mM NaCl, 5 mM β-ME. The appropriate protein fractions were pooled, and the SUMO tag was cleaved with addition of recombinant SUMO hydrolase in a 1:250 (mass/mass) ratio. Due to the presence of the additional glycine residue in front of the starting residue of target protein, the SUMO tag was more flexible and easily cleaved by 10 min incubation on ice. The cleaved protein was dialyzed against 1 L buffer (20 mM Tris-HCl pH 7, 10% w/v glycerol and 5 mM β-ME) overnight at 4 °C and further dialyzed in the fresh buffer for another 2 h to remove all salt. Then the protein was loaded onto a HiTrap SP HP column to subtract the SUMO tag, SUMO hydrolase and other contaminants followed by applying the flowthrough onto an anion exchange Hitrap Q HP column (Cytiva, Marlborough, MA, USA). The captured protein was eluted from the anion exchanged using a 0–40% linear gradient of AEXB buffer (20 mM Tris-HCl pH 7, 1 M NaCl and 5 mM β-ME) over 20 column volumes. After collecting the fractions from anion exchange chromatography, the protein was further concentrated to 30 mg/mL using a 15 mL spin concentrator (10 kDa molecular weight cut-off, Millipore, Burlington, MA, USA) by centrifugation at 4 °C, 4000 rpm and then flash frozen in liquid nitrogen and stored in −80 °C.

The E. coli LeuRS mutant (pETRUK-EcLeuRS-M536G) was produced using the similar quick-change mutagenesis protocol as described above by utilizing the previously reported pETRUK-EcLeuRS[37] construct as the PCR template. The expression and purification of the E. coli LeuRS variant was carried out as for wild-type E. coli LeuRS as discussed in our previous work[37,58].

**Crystallization of wild-type and mutant NgLeuRSs.** Prior to crystallization the wild-type and NgLeuRS mutants were further purified by size exclusion chromatography on a Superdex 200 column (Cytiva, Marlborough, MA, USA) equilibrated in 10 mM Tris-HCl pH 7, 100 mM NaCl, 2.5 mM β-ME. Pooled fractions were concentrated to 10 mg/mL using a microcentrifuge concentrator. For all NgLeuRS variants high quality diffracting crystals were obtained by hanging drop vapor diffusion, in a iterative process using a previously described crystallization condition with the assistance of seeding[37]. Briefly, the SEC purified wild type, E169G and L550G NgLeuRS proteins were mixed with 100 mM Bis-tris propane-HCl pH 8.5, 100 mM MgCl2, 20% (w/v) PEG 3350 and a crystal seed stock in a 0.75:1.0:0.25 (v/v/v) ratio. The seed stock was prepared from crystals of the same construct generated in early rounds of crystallization, that were crushed and diluted in the same precipitant solution. For the NgLeuRS-L550A construct the best crystals were obtained when the pH of the precipitant solution was adjusted to 100 mM Bis-tris propane-HCl pH 8.0, 100 mM MgCl2, 20% (w/v) PEG 3350.

The various NgLeuRS complexes were obtained by transferring apo crystals to a new solution containing the appropriate substrate. NgLeuRS·Leu was obtained by soaking suitable crystals with 10 mM Leu diluted in the cryo-condition consisting of 100 mM Bis-tris propane-HCl pH 8.5, 100 mM MgCl2, 20% (w/v) PEG 3350, 22% (v/v) ethylene glycol; NgLeuRS·ATP·leucinol was obtained by soaking suitable crystals with 5 mM L-leucinol and 5 mM ATP simultaneously in the cryo-condition; NgLeuRS·ATP conformation 1 was obtained by soaking suitable crystals with 10 mM ATP whereas NgLeuRS·ATP conformation 2 was obtained by co-crystallizing NgLeuRS with 10 mM ATP using the above conditions. NgLeuRS·Leu-AMP was obtained by soaking suitable crystals with 5 mM Leu and 5 mM ATP simultaneously in the cryo-condition.

The substrate and intermediate bound complexes of the NgLeuRS-L550G and E169G mutants were also obtained by soaking suitable crystals of the appropriate mutant protein. For all three mutants the ATP·leucinol complex was generated by transferring crystals to a solution of 5 mM ATP and 5 mM L-leucinol diluted in 100 mM Bis-tris propane-HCl pH 8.5, 100 mM MgCl2, 20% (w/v) PEG 3350, 22% (v/v) ethylene glycol. The Leu-AMP complex was achieved by soaking crystals with 5 mM ATP and 5 mM Leu diluted in the same cryo-protection solution. For the NgLeuRS-L550A crystals the ligand-bound complexes were generated by soaking in solutions with the same concentrations of substrate diluted in the lower pH 100 mM Bis-tris propane-HCl pH 8.0, 100 mM MgCl2, 20% (w/v) PEG 3350, 22% (v/v) ethylene glycol cryo-protection. All crystals were soaked for 2 h with the appropriate substrate and subsequently caught in a mounted cryo-loop and flash cooled in liquid nitrogen.

**Data collection and structure determination.** Crystal diffraction data were collected at 100 K on different beamlines at the synchrotron facilities ESRF (Grenoble, France) and Soleil (Paris, France) using a standard data collection strategy. All the data were processed using the autoPROC package[59]. The structures of NgLeuRS complexes in the different catalytic states were initially solved by molecular replacement using Phaser[60] employing the apoenzyme structure (the protein chain plus a zinc ion) from our previously published structure of the same enzyme bound to an intermediate analog[37]. Iterative improvement of the structural models was carried out by multiple rounds of manual correction in Coot[61] and refinement in Phenix.refine[62]. After several rounds of refinement, TLS modeling using groups automatically defined by the Phenix package was applied, and in the final refinement step target weights for stereochemical restraints were automatically optimized. The quality of the final models was validated with wwPDB validation server (https://validate-rcsb-2.wwpdb.org/). Data collection, processing and refinement statistics are summarized in Supplementary Data 1.

All the molecular graphics were generated, and structural comparisons were performed in Pymol (version 2.0.4) and their protein-ligand interactions were analyzed in Schrödinger (release 2021-2).

**Standard kinetic methodology.** [32P]-tRNA[Leu] was prepared as described previously[18,19,63]. The 3′-terminal adenosine of purified Ec-tRNA[Leu] was labeled using tRNA nucleotidyltransferase to exchange A76 of tRNA[Leu] with [α-32P]-ATP. Acceptor activity after labeling was 69 ± 2%, and all rate constants were corrected by this factor to account for the proportion of the functional [32P]-tRNA[Leu]. Ec-tRNA[Leu] was renatured before use by incubation for 3 min at 80 °C, following addition of pre-heated MgCl2 in a final concentration of 10 mM and slowly cooling to ambient temperature.

Amino acid activation was assayed by ATP-PPi exchange, in 100 mM HEPES-KOH, pH 7.5, 30 mM MgCl2, 150 mM KCl, 0.1 mg/mL bovine serum albumin (BSA), 5 mM dithiothreitol (DTT) and 1 mM [32P]-PPi (0.2–0.4 Ci/mol) using 10–20 nM enzyme at 37 °C. For determining steady-state kinetic parameters, Leu was present at 1 mM and the concentration of ATP was varied between $0.1 \times K_M$ and $5–10 \times K_M$, or ATP was present at 25 mM and the concentration of Leu was varied between $0.1 \times K_M$ and $10 \times K_M$. Reactions were started by addition of substrate and stopped in quench solution (750 mM NaOAc pH 4.5 and 0.1 % w/v SDS). Separation of [32P]-ATP and [32P]-PPi was performed by thin-layer chromatography (TLC) in buffer containing 750 mM KH2PO4 pH 3.5 and 4 M urea. Quantization was performed by phosphorimaging using Typhoon Phosphorimager (GE Healthcare) and quantified using ImageQuant as described[64]. Kinetic parameters ($K_M$ and $k_{cat}$) were determined by fitting the data directly to the Michaelis-Menten equation using GraphPad Prism (version 6.0).

The isolated transfer step was assayed in 100 mM HEPES-KOH, pH 7.5, 10 mM MgCl2, 150 mM KCl, 0.1 mg/mL BSA, 5 mM DTT, 8 mM ATP, 0.008 U/μL inorganic pyrophosphatase, 5 mM Leu, 1 μM [32P]-tRNA[Leu], 10–20 μM enzyme at 37 °C. The LeuRS:Leu-AMP complex was preformed in situ by incubating enzyme (20–40 μM) with ATP and Leu for 5 min at 37 °C and mixed with the equal volume of renatured [32P]-tRNA[Leu] (2 μM) using rapid chemical quench instrument (RQF3, KinTek Corp). The LeuRS·Leu-AMP to tRNA ratio higher than one ensures single-turnover conditions. The reaction mixtures were quenched in 600 mM NaOAc, pH 4.5 and 0.1 % (w/v) SDS. [32P]-tRNA[Leu] was further hydrolyzed with ≥ 0.01 U/μL P1 nuclease in 300 mM NaOAc pH 5.0 and 0.15 mM ZnCl2 for 1 h at room temperature to degrade the terminal adenosine of tRNA. Aminoacylated [32P]-AMP was separated from [32P]-AMP by TLC[65] in 100 mM NaOAc and 5% (v/v) HOAc. Kinetic parameter ($k_{trans}$) was determined by fitting the data to a single exponential equation $Y = Y_0 + A \times e^{-k_{trans} \times t}$, where $Y_0$ is the y intercept, $A$ is the amplitude, $k_{trans}$ is the apparent transfer rate constant, and $t$ is the time.

Aminoacylation assays of the EcLeuRS and NgLeuRS variants were performed at 37 °C in reaction buffer containing 100 mM HEPES-KOH pH 7.5, 150 mM KCl, 20 mM MgCl2, 1 mM DTT, 8 mM ATP, 5 mM Leu, 0.1 mg/mL BSA, 0.004 U/μL inorganic pyrophosphatase, up to 40 μM active [32P]-tRNA[Leu], 10 nM enzyme. Steady-state (multi-turnover) conditions were ensured by surplus of substrates over the enzyme. In this assay two-step aminoacylation reaction comprising both the activation and the transfer step is followed. Reactions were started by addition of Leu and stopped in quench solution (750 mM NaOAc pH 4.5, 0.1% w/v SDS). The quenched reactions were further treated with P1 nuclease and separated by TLC. Apparent rate constant ($k_{obs}$) was calculated from the slope of product formation in time divided by the enzyme concentration. Since saturating concentrations of all substrates were used, apparent rate constant closely resembles $k_{cat}$.

**Statistics and reproducibility.** Analysis of the data was performed using GraphPad Prism version 6.0 (San Diego, CA, USA). All values are presented as the means ± standard error of the means. Experiments were performed in triplicates.

**Reporting summary.** Further information on research design is available in the Nature Research Reporting Summary linked to this article.

## Data availability

Atomic coordinates and structure factors have been deposited in the Protein Data Bank (PDB) with the following accession codes: 7NU4 (NgLeuRS), 7NU5 (NgLeuRS in complex Leu), 7NU6 (NgLeuRS in complex with ATP in conformation 1), 7NU7 (NgLeuRS in complex with ATP in conformation 2), 7NU8 (NgLeuRS in complex with Leu-AMP), 7NU9 (NgLeuRS in complex with ATP and L-leucinol), 7NTY (NgLeuRS-L550A), 7NTZ (NgLeuRS-L550A in complex with Leu-AMP), 7NU0 (NgLeuRS-L550A in complex with ATP and L-leucinol), 7NUA (NgLeuRS-L550G), 7NUB (NgLeuRS-L550G in complex with Leu-AMP), 7NUC (NgLeuRS-L550G in complex with ATP and L-leucinol), 7NU1 (NgLeuRS-E169G), 7NU2 (NgLeuRS-E169G in complex with Leu-AMP), 7NU3 (NgLeuRS-E169G in complex with ATP and L-leucinol). The data used for the generation of the plots in Supplementary Fig. 4, from which the enzymatic kinetic parameters presented in Table 1 were determined, are provided in the Supplementary Data 2 file.

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

## Acknowledgements

This work was supported by the Research Fund Flanders [Fonds voor Wetenschappelijk Onderzoek, G077814N to S.V.S. and A.V.A., G0A4616N to S.D.W. and A.V.A.], the KU Leuven Research Fund [3M14022 to S.D.W. and A.V.A.], Croatian Science Foundation [Grant IP-2016-06-6272 to I.G.-S.] and the CSC scholarship to L.P. We further greatly appreciate the support from the beamline scientists at the PROXIMA 1 and PROXIMA 2 (Soleil Synchrotron, France), and ID23-1 and MASSIF-3 (ESRF, France) beamlines.

## Author contributions

L.P. and S.D.W. conceived the project and designed most of the experiments. L.P. cloned, expressed and purified all NgLeuRS and EcLeuRS variants for crystallization and biochemical studies. L.P produced and harvested the different NgLeuRS crystal complexes, collected the X-ray diffraction datasets, determined the structures and performed the bioinformatic analyses with the assistance of S.D.W. All efforts by L.P. were guided by S.D.W, S.V.S and A.V.A. V.Z. and I.G.-S. designed and performed enzymatic kinetic studies of the various LeuRS mutants and analyzed and interpreted the data. L.P. and S.D.W. wrote the manuscript with contributions from V.Z. and I.G.-S. All authors discussed the results and were involved in editing of the final version of the manuscript.

## Competing interests

L.P., V.Z., S.V.S., A.V.A., and I.G.-S. declare no competing interests. S.D.W. is an employee of Pledge Therapeutics B.V. and therefore declares a potential competing interest.
