## [Peer Review File · Communications Biology]

Reviewers' comments:

Reviewer #1 (Remarks to the Author):

The work describes several structural snapshots of the LeuRS enzyme at various stages of the adenylation process, combined with biochemical characterization of the WT protein and structure-based designed mutants. The authors observe a new conformational state associated with a peptide switch that appears to be driving the protein rearrangement between different stages of the overall reaction.

In general, I find the work to be solid, with a detailed description of structural intricacies, but most likely accessible only to the experts in the field of amino acyl tRNA synthetases.

Some specific comments that could improve the manuscript are below:

1) Introduction would benefit from a broader description of class I aaRSs, instead of placing LeuRS in the context of a broader family of adenylate forming enzymes. It might be emphasized (with references) that adenylation is tRNA-independent, unlike in some other aaRSs.

2) I am under the impression that the refinement protocol is not optimal. I see R_{free} as somewhat high for the obtained resolution and in some instances the gap between R_{work} and R_{free} is large too. While the models do not seem to carry any major errors, I would suggest reevaluating the approach. For example, the text does not mention whether TLS was used, and if so, how. Also, maybe weight optimization could be tried. Some side chains could be possibly flipped, per validation reports. I could not evaluate these myself as I haven't had access to the data.

3) Validation reports point to more sequence discrepancies than one expects from the paper. While the "conflict" residue (508) may result just from the wrong entry used as a reference, the "engineered mutation" implies the change has been deliberate. This refers to 454Asp to Asn mutation that is not discussed in the text.

4) The first reaction carried out by the enzyme (adenylation) is often referred to as the "first step of catalysis", which does not seem to be correct as the text does not go into mechanistic details of the actual catalysis and its individual steps. Perhaps a different wording could be found.

5) Line 273: Unless I missed something, there is no actual evidence WT EcLeuRS undergoes a peptide flip and M536G does not. In addition, there are sequences (M. mycoides) that natively have Gly in this position. Can anything be said about these proteins?

6) 329: "Our biochemical data show that this peptide-plane flip is not necessary for the first step of catalysis, but once the intermediate is formed it flips back to its favored position, inducing the repositioning of the CP1 hairpin"

I don't see how the first half of the sentence connects with the second half. Plus, as I stated earlier, I have a problem with "catalysis" here. In addition, while the flip per se is not necessary in the first reaction, it is needed for L550 to avoid clashes.

7) Figure 1. Anti-codon binding domain and the C-terminal domain colors look similar. Changing the color of either domain might be helpful. Also, in panel (a) the background could be brighter.

Minor aspects

Abstract: Please, provide the organism name

As the paper follows American English, I'd suggest using "homolog", "analog" etc. vs.

"Homologue", "analogue". Or at least be consistent, currently both forms are used.

Line 74: should be "acylated" not "acetylated"

100: Holoenzyme? I would explicitly say here that the model was a complex with the intermediate analog (as stated in the Methods)

115: Not clear what a "holoenzyme" is in this case. I think it's actually meant to be the "apo" here.

117: amino group (not amine)

120: What "apo state" is referred to here? I guess NgLeuRS, but with "holoenzyme" used above I got confused.

151: Assuming "transition" refers to aminoacylation, 6Q89 also represents a pre-transition state. Perhaps a different term could be used to differentiate? It might help if some terms were defined early in the text.

Line 153, 163, 175: It is Mg²⁺ ion that is coordinated by ATP (and/or water molecules), not vice versa. Similarly, Fig. 2 should read "coordinating water molecules"

Line 166: carboxylate oxygen atom of Leu

170: structural water molecules

175: three structured water molecules

176: add a comma before "suggesting that"

Line 179, Leu-AMP – any differences with 6Q89?

201: Ramachandran (capital R)

292, 325: carboxylate oxygen atom

350: amino acid residues

585: there is an extra bracket before "7NU2"

Unless it's a journal requirement, I'd use small capital L for stereochemistry

Methods: Please, provide full description of the buffers, Hepes, Tris – Na? K?

Crystallization: Not clear if the mutants were also purified by SEC. Please, provide the crystallization conditions.

Fig. 2. These images need to be bigger and higher resolution. Perhaps it's the pdf conversion that made the figure both smaller and of lower resolution. I tried to zoom in on the screen to see it better, but then the lack of resolution was quite striking.

Leu-AMP is not described in the legend.

Fig. 3: is this a weighted map?

Extended Fig. 1a – the focus is on the map quality, so I'd present here only the ligand and its map, the protein cartoon is not necessary here.

Extended Fig. 3 – EcLeuRS-WT is missing the error

Table 1: I find the notion of parameters and their units awkward, please use brackets instead of /
Suppl. Table 1: The statistics table needs to be cleaned up:

1. Space group notation

2. Number of significant digits needs to be identical for the same parameters. In many cases, there should be also fewer significant digits as the numbers imply unrealistic precision.

Reviewer #2 (Remarks to the Author):

Please see the attached PDF.

Copied from PDF by Editor (may contain errors in converting font so also see attached PDF):

In this paper, the authors have solved crystal structures of *N. gonorrhoeae* leucyl-tRNA synthetase (LeuRS) for all enzymatic states during the activation of leucine by ATP to form leucyl-adenylate (Leu-AMP). They compared these structures and with biochemical analyses and mutant structural studies, they proposed the role of a cycle of dramatic conformational changes, involving multiple domains and energetically unfavorable peptide-plane flip observed in the active site in the reaction. The authors have been the specialists in this research field and their results obtained are extensive and clear. It should be highly noted that their strategy is quite logical and the process attaining to the conclusion is scientifically very sound. I recommend the paper should be published in *Communications Biology* after proper revision regarding the following points:

(1) Although the authors showed both structures of LeuRS-Leu and LeuRS-ATP states, does the former really reflect the situation *in vivo*? The K_m values shown in Table 1 suggest that ATP is better binder to LeuRS than Leu, and I am wondering if the conformational change by the binding of ATP may cause the sequential uptake of Leu. The authors should discuss the point in considering the K_m values and *in vivo* concentration of ATP and Leu.

(2) Superposition with the Leu-bound structure shows that the carboxylate oxygen of Leu is located in the required position to allow nucleophilic attack on the α -phosphate of ATP at a (O-P) distance of 2.9 Å. However, carboxylate ion is usually delocalized and nucleophilicity is reduced compared to the localized form. Therefore, for efficient nucleophilic reaction, proper residue(s) on LeuRS could contribute the localization of the carboxylate ion required. It would be even better if the authors could comment on this point from the structure obtained.

(3) The title of Fig. 4 legend is misleading because Fig. 4(a) did not show the results of L550A/G and E169G mutants. In addition, the labels K_M (Leu) and k_{cat} in the upper panel seem to be hard to understand. The authors should reconsider the points.

(4) In Table 1, LeuRS-L550G showed only a 2-fold decrease in the k_{cat}/K_m for Leu activation relative to the WT (first step). However, in the second step (leucine transfer step to tRNA), k_{cat}/K_m value showed about 30-fold drop. In contrast to LeuRS-L550G, the kinetic behaviors in the first and second steps seems to be qualitatively opposite in the case of LeuRS-E169G. It would be better if there was a more sufficient explanation for these differences from the standpoint of their experiments.

(5) Although the authors concluded that their structural characterization of *N. gonorrhoeae* LeuRS provides a high-resolution structural framework providing the first insights into the amino acid activation mechanism of class Ia aaRSs, historically Alan Fersht has performed extensive works on TyrRS (also class I) in the structural standpoints. The authors should cite more his works other than reference 25 and discuss the relationship with their present results.

<Minor points>

Line 69, Line 73: Leu- tRNA -> Leu-tRNA (No space between - and t.)

Reference 24: tRNA^{Leu} -> tRNA^(Leu)

Reference 55: tRNA^{Ala} -> tRNA^(Ala)

Reference 7, 15, 21, 22, 29, 34, 36: Please write using lowercase alphabets, except at the beginning of the title of the papers.

We thank both Reviewers for their very careful and considered analysis of our initial submission. Numerous important points were raised, and we have attempted to address these in the updated article. We believe these critiques and suggestions have helped strengthen the manuscript by clarifying some ambiguity in the description. In the subsequent pages we have included the point-by-point response to the Reviewers. Initial comments from each reviewer are shown in blue text and the response/correction in black.

Reviewer #1 (Remarks to the Author):

The work describes several structural snapshots of the LeuRS enzyme at various stages of the adenylation process, combined with biochemical characterization of the WT protein and structure-based designed mutants. The authors observe a new conformational state associated with a peptide switch that appears to be driving the protein rearrangement between different stages of the overall reaction.

In general, I find the work to be solid, with a detailed description of structural intricacies, but most likely accessible only to the experts in the field of aminoacyl-tRNA synthetases.

Some specific comments that could improve the manuscript are below:

1) Introduction would benefit from a broader description of class I aaRSs, instead of placing LeuRS in the context of a broader family of adenylate forming enzymes. It might be emphasized (with references) that adenylation is tRNA-independent, unlike in some other aaRSs.

Response: AaRSs are prototypical members of the mechanistically related superfamily of adenylate-forming enzymes. As stated in our opening paragraph of the introduction, this structurally and functionally diverse family all utilize ATP to form an acyl-AMP intermediate, and then transfer the activated acyl group to a second substrate. Most of these enzymes perform the two reactions at the same site, but how these steps are compartmentalized is a fundamental question which needs to be explored. Our intent in the first paragraph was to appeal to the broader scientific community. Members of this superfamily are well known targets for drug development¹ but true exploitation of this enzyme class needs a more thorough understanding of the catalytic steps and how these relate to the complex structural dynamics of these enzymes.

However, we do agree with the reviewer that it is important to place our study in context with efforts that have been previously performed on other class I aaRSs. Staying within the recommended restrictions of the journal we have modified lines 67-75 and lines 86-91. We hope the addition of this useful suggestion will also clarify Point number 5 of Referee #2.

2) I am under the impression that the refinement protocol is not optimal. I see Rfree as somewhat high for the obtained resolution and in some instances the gap between Rwork and Rfree is large too. While the models do not seem to carry any major errors, I would suggest reevaluating the approach. For example, the text does not mention whether TLS was used, and if so, how. Also, maybe weight optimization could be tried. Some side chains could be possibly flipped, per validation reports. I could not evaluate these myself as I haven't had access to the data.

Response: In terms of the potential side-chain flips in the structure, we have carefully inspected the PDB validation reports and the associated structures. Most of the earmarked side chains are located on the surface of the protein structure, exposed to the solvent and with no obvious support for a specific side chain orientation. Even so we have flipped the side chains and redeposited structures. There are however a few cases where the proposed flip does not make sense. In particular Asn55 and His547, which are located in the active site, make essential H-bonds with substrate ATP and Leu and reaction intermediate Leu-AMP, but are often flagged by the PDB validation software even in the ligand bound structures. As these interactions are important for ligand binding (see figure 1.1 and figure 2 in the manuscript), we have kept the side-chain orientation as is. We have uploaded the updated merged PDB validation reports alongside the revised manuscript.

Figure 1.1. Protein-ligand interactions of NgLeuRS·Leu-AMP complex. The protein backbone is shown as cartoon representation while the bound ligand and the interacting residues His547 and Asn55 are shown as sticks. H-bonds are shown as black dashed lines.

Concerning the R-values, we agree with the reviewer that these are quite high for the deposited structures. Prior to submission we did test a variety of refinement strategies and compared different software (Phenix, Refmac and Buster) without any discernible improvement in the latter case. However, a comparison of our models versus the PDB using the r-factor statistics tool in the Phenix package shows that overall, they fall within the majority of structures in the same resolution range. In the figure below we show the results of such an analysis for the Leu bound wild-type structure (NgLeuRS·Leu).

Figure 1.2: R-factor value distribution of deposited PDBs refined to a resolution of 2.56 - 2.6 Å. Grey arrow corresponds to statistics calculated for the Leu soaked NgLeuRS structure (Rwork=0.202, Rfree=0.269, Rfree-Rwork=0.067)

The disparity observed is likely a combination of observed anisotropic diffraction, a reflection of the elongated protein shape of NgLeuRS, and the dynamics of this enzyme in the crystalline state. In the latter case, recognizing that NgLeuRS is a multi-domain protein, we observed considerable flexibility of the C-terminal and leucine-specific domains, which account for ~14% of the total protein, between our different crystal structures. As mentioned in the opening paragraph of the results, we could only fully trace NgLeuRS in complex with ATP and leucinol. In the other structures there is clear evidence for the presence of these domains in the calculated map, but we have been conservative in our attempts to model these regions to minimize RSRZ outliers which is also an important criterium during PDB validation. A better approach to modelling the data, likely yielding lower R values, would probably be best served by using ensemble modeling. However, as the quality of the calculated electron density map in the regions under investigation allowed for unambiguous modelling of the protein and ligands, we chose to employ the standard “static” snapshot for refinement which employed both a regular isotropic description of the B-factors of individual atoms as well as utilized TLS groups.

We have updated the “Data collection and structure determination” section (lines 537-549) of the methods to include this refinement protocol and included additional references to the appropriate software used:

“Crystal diffraction data were collected at 100 K on different beamlines at the synchrotron facilities ESRF (Grenoble, France) and Soleil (Paris, France) using a standard data collection strategy. All the data were processed using the autoPROC package². The structures of NgLeuRS complexes in the different catalytic states were initially solved by molecular replacement using Phaser³ employing the apoenzyme structure (the protein chain plus a zinc ion) from our previously published structure of the same enzyme bound to an intermediate analog⁴. Iterative improvement of the structural models was carried out by multiple rounds of manual correction in Coot⁵ and refinement in Phenix.refine⁶. After several rounds of refinement, TLS modeling using groups automatically defined by the Phenix package was applied, and in the final refinement step target weights for stereochemical restraints were automatically optimized. The quality of the final models was validated with wwPDB validation server (<https://validate-rcsb.wwpdb.org>). Data collection, processing and refinement statistics are summarized in Supplementary table 1.”

3) Validation reports point to more sequence discrepancies that one expects from the paper. While the “conflict” residue (508) may result just from the wrong entry used as a reference, the “engineered mutation” implies the change has been deliberate. This refers to 454Asp to Asn mutation that is not discussed in the text.

Response: The DNA sequence encoding NgLeuRS was amplified from the genomic DNA of *N. gonorrhoeae* ATCC 49226 isolated from a patient. The sequence integrity of the working construct was further confirmed by sanger sequencing. A manual BLAST of our protein sequence within Uniprot database showed B4RNT1 (redundant with A0A5K1KQ39) having 99.8% sequence identity. Residues 454 and 508 are therefore possibly natural mutations of the enzyme in this *N. gonorrhoeae* strain or may have resulted from PCR mutation. In either case they have no-role in the described peptide-plane flip phenomenon. We have requested that the PDB annotate both residues as “conflict” to avoid the understandable confusion raised by the reviewer.

4) The first reaction carried out by the enzyme (adenylation) is often referred to as the “first step of catalysis”, which does not seem to be correct as the text does not go into mechanistic details of the actual catalysis and its individual steps. Perhaps a different wording could be found.

Response: We agree with the reviewer that the term “first step of catalysis” may be misleading. Therefore, in the revised version we use terms the first and the second step of aminoacylation or the amino acid activation step and aminoacyl transfer step or abbreviated, the activation and the transfer step.

5) Line 273: Unless I missed something, there is no actual evidence WT EcLeuRS undergoes a peptide flip and

M536G does not. In addition, there are sequences (*M. mycoides*) that natively have Gly in this position. Can anything be said about these proteins?

Response: Our kinetic data (Table 1) do show that *E. coli* LeuRS-M536 is relevant. Mutation of this residue is similar to the NgLeuRS-L550G in that the transfer step is more affected than the activation step. However, the reduction in the rate of the former is only 4-fold compared to the 30-fold reduction seen for NgLeuRS. We have carefully examined available structures of EcLeuRS and comparison of the structure of an “apo” active site versus the intermediate analog bound state do show that the backbone in this region in the former case is distorted at M536 (see Figure 1.4 A and B below). Unfortunately, to date EcLeuRS has only been crystallized with and without substrates in the presence of tRNA^{Leu}, and the available structures with an intermediate analog – equivalent to our Leu-AMP - was achieved by co-crystallization. Therefore, we are not certain whether the observed state is a crystallographic artefact or truly reflecting the structure during the catalytic cycle. Similar distortions of the N-terminus of the equivalent helix are observed in the LeuRS from *Thermus thermophilus* (PDB IDs: 1OBH and 1H3N) and *Mycoplasma mobile* (PDB ID: 3ZIU) where the CP1 hairpin is always found in closed conformation. Like the EcLeuRS models, these structures were obtained by co-crystallization and thus also must be considered with some caution. In conclusion, there is structural evidence that this highly conserved region of LeuRS is distorted in *E. coli* and other bacterial LeuRS homologs, but we have been conservative in extending this analysis in our report, as these earlier published structures do not demonstrate that the enzyme is active in the crystal. Accepting this is a weakness in translating our findings to other bacterial leuRS we have changed the language of line 285 to be:

“Taken together, M536G mutation in EcLeuRS produced similar but less disruptive effects as compared with L550G in NgLeuRS, suggesting that the use of the peptide-plane flip for catalysis could be shared among other bacterial LeuRSs (Table 1)”

In terms of the second part of the reviewer’s comment. For *M. mycoides* LeuRS, and other species with a glycine at the L550 position, we do recognize that this is important question. Examination of the extensive sequences available show at least 5 different architectures across all kingdoms. Focusing on the catalytic region and the associated domains, these vary in terms of positioning the CP1 hairpin, and the presence or absence of additional domains (See figure 1C below). There are principally two subgroups of LeuRSs that containing the equivalent of the G/A550 mutant. The major group, which also contains a shorter aspartate residue in a position equivalent to E169 in NgLeuRS, the majority have an additional domain upstream of the CP1 hairpin which may affect the movement of the latter. The other subgroup has G550 mutant (the equivalent residue to E169 can be a Glu or Asp) but lack the LS domain. For mycoplasma genera, the sequence varies dramatically between species. Some have degenerate/truncated leucine-specific and editing domains as ub in *M. mobile*, while some species have regular editing domain as seen for bacterial like *Mycoplasma neurolyticum* LeuRS (WP_129720014.1). Compartmentalization of the catalytic steps in *Mycoplasma* is therefore likely to be different depending on the species. As mentioned above, an examination of the *M. mobile* structure does show distortion of the L550-helix but we are lacking enough structures of the different catalytic states, and of the different LeuRS members, to confidently extrapolate what we observe with NgLeuRS to the family as a whole.

Crucially, our data does suggest that the peptide-plane flip is important in prototypical bacterial/mitochondrial enzymes but, as we argue in our paper, how different members of this extremely large superfamily of enzymes partition the substrate adenylation step from the transfer step has, to the best of our knowledge, not previously been investigated. We are certain there will be a large variety of mechanisms and believe that our work we stimulate more studies in this fundamental area of enzymatic catalysis.

Figure 1.3. Examination of the M536-containing α -helix of different catalytic states of EcLeuRS. (A) Structural superposition of EcLeuRS in complex with tRNA in the aminoacylation state (PDB ID: 4AQ7) and post-transfer editing state (PDB ID: 4AS1). M536 is distorted in post-transfer state. (B) Ramachandran plot of M536 and I535 in two different catalytic states of EcLeuRS. M536 residues in both structures are positioned in the right-handed alpha helix region and both are Ramachandra unfavorable. (C) Domain structures of the representatives of LeuRS homologs

6) 329: “Our biochemical data show that this peptide-plane flip is not necessary for the first step of catalysis, but once the intermediate is formed it flips back to its favored position, inducing the repositioning of the CP1 hairpin” I don’t see how the first half of the sentence connects with the second half. Plus, as I stated earlier, I have a problem with “catalysis” here. In addition, while the flip per se is not necessary in the first reaction, it is needed for L550 to avoid clashes.

Response: We agree with the reviewer that the flip is needed to avoid the clash. We have reformulated the sentence (Line 347) to avoid other confusions as well:

“Our biochemical data show that in the absence of steric constraints the peptide-plane flip is not necessary for the amino acid activation. Instead, the flip back to its favored position once the intermediate is formed, enables the aminoacyl transfer step by promoting the repositioning of the CP1 hairpin.”

7) Figure 1. Anti-codon binding domain and the C-terminal domain colors look similar. Changing the color of either domain might be helpful. Also, in panel (a) the background could be brighter.

Response: We have revised all panels of Figure 1, and modified the associated legend, based on the suggestions of the reviewer. The new coloring was also applied to Figure 2 (top right panel) and Figure 5.

Fig. 1 | Biochemical and structural boundaries of *N. gonorrhoeae* LeuRS. (a) The enzymatic steps catalyzed by LeuRS. The first two stages occur in the aminoacylation domain (green box) while the post-transfer editing of mis-charged Nva-tRNA^{Leu} is processed in the editing domain (cyan box). Nva is the abbreviation of non-canonical amino acid L-norvaline. (b) The domain structure of *N. gonorrhoeae* LeuRS and (c) its corresponding cartoon representation. The aminoacylation domain (green) is split by multiple insertions: connective polypeptides 1 (CP1 hairpin, purple) and 2 (CP2, slate); the editing domain (cyan); the zinc binding domain (Zn2, yellow) and the leucine specific domain (LS, orange). The aminoacylation domain is followed by the anti-codon binding domain (ACBD, red) and the C-terminal domain (CTD, brown). A zinc ion is shown as a grey sphere.

8) Minor aspects

Abstract: Please, provide the organism name

Response: The organism's name has been modified the appropriate sentence in the abstract; lines 45-46:

“We have solved crystal structures for all enzymatic states of *Neisseria gonorrhoeae* LeuRS during Leu-AMP formation.”

As the paper follows American English, I'd suggest using “homolog”, “analog” etc. vs. “Homologue”, “analogue”. Or at least be consistent, currently both forms are used.

Response: These have been checked and revised to be consistent throughout the text.

Line 74: should be “acylated” not “acetylated”

Response: Thank you for spotting this, it has been corrected.

Line 117: amino group (not amine)

Response: Revised.

Line 100: Holoenzyme? I would explicitly say here that the model was a complex with the intermediate analog (as stated in the Methods)

Line 115: Not clear what a “holoenzyme” is in this case. I think it’s actually meant to be the “apo” here.

Line 120: What “apo state” is referred to here? I guess NgLeuRS, but with “holoenzyme” used above I got confused.

Response: As the protein contains a bound zinc ion in the Zn2 domain we initially used the word holoenzyme to represent this metal ion bound structure. We recognize from the raised points that we have switched interchangeably between the word holoenzyme and apoenzyme throughout the text which has led to understandable confusion. A survey of the literature does suggest that the phrase apoenzyme is much more frequently used when discussing this enzyme class and therefore we have edited the text to use this term exclusively. In terms of line 100 the initial structure solution was performed using the protein and zinc atom ion extracted from coordinates of the protein in complex with a non-hydrolysable analog of Leu-AMP (PDB ID: 6Q89). We have clarified this latter point in the methods section (see highlighted text in the response to the Reviewer 1, point # 2).

Line 151: Assuming “transition” refers to aminoacylation, 6Q89 also represents a pre-transition state. Perhaps a different term could be used to differentiate? It might help if some terms were defined early in the text.

Response: We use the term transition state to mean the pentavalent stabilized state where the carboxylic oxygen atom has attacked the α -phosphate of ATP but the pyrophosphate group has not left. Our NgLeuRS·ATP·leucine captures the enzyme just before the pentacoordinate phosphorous is formed. The Leu-AMP and the sulfamoyl linked analog in 6Q89 show the intermediate following resolution of this temporally short lived state.

In the second paragraph (line 69) of the introduction, we have added a statement with appropriate references to help clarify this:

“LeuRS recognizes leucine (Leu) and ATP to form a leucyl-adenylate (Leu-AMP) intermediate, a reaction that proceeds via a pentavalent transition state⁷⁻⁹.”

Line 153, 163, 175: It is Mg²⁺ ion that is coordinated by ATP (and/or water molecules), not vice versa. Similarly, Fig. 2 should read “coordinating water molecules”

Response: We have modified the lines to the following:

Line 163: “The triphosphate group of ATP is fully ordered, **coordinating** a Mg²⁺ ion and making extensive interactions with residues of the class I signature HIGH motif which has the sequence of ⁴⁹HMGH⁵² in NgLeuRS, with the KMSKS loop and the CP1 hairpin”

Line 173: “In addition, the α , β , γ -phosphates of ATP and three additional ordered water molecules **coordinate** a Mg²⁺ ion, forming an ideal octahedral geometry with an average distance around 2 Å between Mg²⁺ and the contacting atoms (Fig. 2 and Extended Data Fig. 1d).”

Line 185: “In addition, superposition of *Geobacillus stearothermophilus* TrpRS·ATP·tryptophanamide and our NgLeuRS ternary structures based on the adenosine moiety of ATP (RMSD 0.09 Å for all ATP atoms) showed that the triphosphate group in both structures shares a very similar extended conformation **with the triphosphate of ATP, equivalently coordinating with a Mg²⁺ ion.**”

Line 166: carboxylate oxygen atom of Leu

Response: Revised

170: structural water molecules

Response: Revised.

175: three structured water molecules

Response: Revised

176: add a comma before “suggesting that”

Response: Revised

Line 179, Leu-AMP – any differences with 6Q89?

Response: 6Q89 is the crystal structure of NgLeuRS in complex with leucyl-sulfamoyl adenosine (LSA) a non-hydrolysable analog of the reaction intermediate Leu-AMP (See figure below). Similar to our structures, the 6Q89 structure was determined from apo crystals soaked with LSA using the same crystallization conditions. The two protein structures can be fully superimposed with overall RMSD of 0.175 Å (713 Ca atoms). LSA and Leu-AMP bind in the same manner in LeuRS active site and make the same interactions with surrounding protein residues, and the positions of the KMSKS loop, the CP1 hairpin and the editing domain are equivalent. We have intentionally chosen not to compare these structures in the manuscript, as the non-hydrolysable analog alone does not prove that the enzyme is catalytically active in the crystal which is essential for the current study.

201: Ramachandran (capital R)

Response: In line 201 (215 in the revised manuscript), “ramachandran plots” was corrected as “Ramachandran plots”.

292, 325: carboxylate oxygen atom

Response: “the carboxylate oxygen of Leu” in line 292 and 325 (307 and 344 in the revised manuscript) were revised as “the carboxylate oxygen atom of Leu”

350: amino acid residues

Response: In line 350 (370 in the revised manuscript), “two conserved protein residues” was revised as “two conserved amino acid residues”.

585: there is an extra bracket before “7NU2”

Response: Revised.

Unless it's a journal requirement, I'd use small capital L for stereochemistry

Response: The small capital L ("L") was applied for stereochemistry through the entire manuscript according to your suggestion.

Methods: Please, provide full description of the buffers, Hepes, Tris – Na? K?

Response: Full description of the buffers, HEPES, Tris and Bis-tris propane were added to the methods section of the manuscript.

Crystallization: Not clear if the mutants were also purified by SEC. Please, provide the crystallization conditions.

Response: The mutants were also purified by SEC and crystallized using the same conditions. To help clarify this we have simplified and updated this section of the manuscript (lines 501-535):

“Prior to crystallization the wild type and NgLeuRS mutants were further purified by size exclusion chromatography on a Superdex 200 column (Cytiva, Marlborough, MA, USA) equilibrated in 10 mM Tris-HCl pH 7, 100 mM NaCl, 2.5 mM β -ME. Pooled fractions were concentrated to 10 mg/mL using a microcentrifuge concentrator. For all NgLeuRS variants high quality diffracting crystals were obtained by hanging drop vapor diffusion, in an iterative process using a previously described crystallization condition with the assistance of seeding¹⁰. Briefly, the SEC purified wild type, E169G and L550G NgLeuRS proteins were mixed with 100 mM Bis-tris propane-HCl pH 8.5, 100 mM MgCl₂, 20% (w/v) PEG 3350 and a crystal seed stock in a 0.75:1.0:0.25 (v/v/v) ratio. The seed stock was prepared from crystals of the same construct generated in early rounds of crystallization, that were crushed and diluted in the same precipitant solution. For the NgLeuRS-L550A construct the best crystals were obtained when the pH of the precipitant solution was adjusted to 100 mM Bis-tris propane-HCl pH 8.0, 100 mM MgCl₂, 20% (w/v) PEG 3350.

The various NgLeuRS complexes were obtained by transferring apo crystals to a new solution containing the appropriate substrate. NgLeuRS·Leu was obtained by soaking suitable crystals with 10 mM Leu diluted in the cryo-condition consisting of 100 mM Bis-tris propane-HCl pH 8.5, 100 mM MgCl₂, 20% (w/v) PEG3350, 22% (v/v) ethylene glycol; NgLeuRS·ATP·leucinol was obtained by soaking suitable crystals with 5 mM L-leucinol and 5 mM ATP simultaneously in the cryo-condition; NgLeuRS·ATP conformation 1 was obtained by soaking suitable crystals with 10 mM ATP whereas NgLeuRS·ATP conformation 2 was obtained by co-crystallizing NgLeuRS with 10 mM ATP using the above conditions. NgLeuRS·Leu-AMP was obtained by soaking suitable crystals with 5 mM Leu and 5 mM ATP simultaneously in the cryo-condition.

The substrate and intermediate bound complexes of the NgLeuRS-L550G and E169G mutants were also obtained by soaking suitable crystals of the appropriate apo mutant protein. For all three mutants the ATP·leucinol complex was generated by transferring crystals to a solution of 5 mM ATP and 5 mM L-leucinol diluted in 100 mM Bis-tris propane-HCl pH 8.5, 100 mM MgCl₂, 20% (w/v) PEG3350, 22% (v/v) ethylene glycol. The Leu-AMP complexes was achieved by soaking crystals with 5 mM ATP and 5 mM Leu diluted in the same cryoprotection solution. For the apo NgLeuRS-L550A crystals the ligand bound complexes were generated by soaking in solutions with the same concentrations of substrate diluted in the lower pH 100 mM Bis-tris propane-HCl pH 8.0, 100 mM MgCl₂, 20% (w/v) PEG 3350, 22% (v/v) ethylene glycol cryoprotection. All crystals were soaked for 2 hours with the appropriate substrate and subsequently caught in a mounted cryo-loop and flash cooled in liquid nitrogen”

Fig. 2. These images need to be bigger and higher resolution. Perhaps it's the pdf conversion that made the figure

both smaller and of lower resolution. I tried to zoom in on the screen to see it better, but then the lack of resolution was quite striking.

Leu-AMP is not described in the legend.

Response: The figure was generated at 1200 DPI as should hopefully appear at higher resolution in the final document. The absent identification of Leu-AMP has been added to the legend of figure 2.

Fig. 3: is this a weighted map?

Response: As stated in the legend the aminoacylation domain, CP2, Zn₂, anti-codon binding domain and C-terminal domains of NgLeuRS·ATP·leucinol complex area additionally shown as a surface representation generated in pymol. The purpose of this view was to distinguish between the regions of the protein that can be superposed with low RMSD to the mobile regions, the ribbon diagram on its own being quite busy.

Extended Fig. 1a – the focus is on the map quality, so I'd present here only the ligand and its map, the protein cartoon is not necessary here.

Response: The revised Extended Fig. 1a is shown below.

Extended Data Fig. 1 | NgLeuRS-substrate complexes. (a) Omit maps (grey mesh) of the ligands in the different solved LeuRS-substrate complexes calculated using Phenix.Polder. The reported real-space correlation coefficient (RSCC) of each ligand shows that they are well bound in the active site of NgLeuRS. Two different ATP binding conformations were obtained by co-crystallization and soaking methods. The Polder maps of ligands are countered in a range of 3.5 - 5 σ . (b) Comparison of the active site conformation of the ligand-free and different substrate-bound NgLeuRS structures. The protein backbones are shown as ribbon representations, specific protein residues and substrates are shown as sticks. NgLeuRS, NgLeuRS•Leu, NgLeuRS•ATP (obtained by co-crystallization), NgLeuRS•ATP•leucinol and NgLeuRS•Leu-AMP are colored in slate, grey, purple, magenta and green, respectively. The residues with different rotamers and conformational changes of the KMSKS loop are highlighted. Individual structures were superposed using residues encompassing the aminoacylation, CP2, Zn² and anti-codon binding domains. (c) Superposition of the N-terminus of L550-containing α -helix (residues H547-E565) in the apo and Leu-bound structures. The H bonds in the apo structure are shown as black dashed lines while the H-bond in

NgLeuRS·Leu is shown as a yellow dashed line. (d) Comparison of ligand bound conformations in the active site of the superposed structures of NgLeuRS·Leu and NgLeuRS·ATP·leucinol. The magnesium ion is shown as a grey sphere with surrounding water molecules shown as small red spheres. (e) Structural alignment of NgLeuRS·ATP·leucinol with *Geobacillus stearothermophilus* TrpRS·ATP·tryptophanamide based on the adenine. The carbon atoms of ATP, leucinol and the magnesium ion in the former structure are colored in cyan, salmon and grey, respectively, while in the TrpRS·ATP·tryptophanamide structure, the corresponding carbon atoms of ATP, tryptophanamide and magnesium ion are all green.

Extended Fig. 3 – EcLeuRS-WT is missing the error

Response: Due to significant amounts of the protein and tRNA required for pre-steady state kinetic data obtained by KinTek rapid chemical quench instrument, we have performed this experiment only once. EcLeuRS is extensively characterized in our lab (Cvetesic et al, J. Biol. Chem. 2012, 287(30):25381-94., Cvetesic et al, EMBO J, 2014, 33(15):1639-53) and this experiment confirmed the previous results.

Table 1: I find the notion of parameters and their units awkward, please use brackets instead of /

Response: Revised.

Suppl. Table 1: The statistics table needs to be cleaned up:

1. Space group notation
2. Number of significant digits needs to be identical for the same parameters. In many cases, there should be also fewer significant digits as the numbers imply unrealistic precision.

Response: Thank you for your suggestions. Both points have been addressed in an updated Supplementary table 1.

Reviewer #2 (Remarks to the Author):

In this paper, the authors have solved crystal structures of *N. gonorrhoeae* leucyl-tRNA synthetase (LeuRS) for all enzymatic states during the activation of leucine by ATP to form leucyl-adenylate (Leu-AMP). They compared these structures and with biochemical analyses and mutant structural studies, they proposed the role of a cycle of dramatic conformational changes, involving multiple domains and energetically unfavorable peptide-plane flip observed in the active site in the reaction. The authors have been the specialists in this research field and their results obtained are extensive and clear. It should be highly noted that their strategy is quite logical and the process attaining to the conclusion is scientifically very sound. I recommend the paper should be published in Communications Biology after proper revision regarding the following points:

(1) Although the authors showed both structures of LeuRS-Leu and LeuRS-ATP states, does the former really reflect the situation in vivo? The K_M values shown in Table 1 suggest that ATP is better binder to LeuRS than Leu, and I am wondering if the conformational change by the binding of ATP may cause the sequential uptake of Leu. The authors should discuss the point in considering the K_M values and in vivo concentration of ATP and Leu.

Response: NgLeuRS exhibits the $K_M(\text{Leu})$ of 14.3 μM and $K_M(\text{ATP})$ of 2.5 mM or 2500 μM , the latter being 174-fold higher, suggesting a two orders of magnitude lower affinity for ATP. It has been reported that in exponentially grown glucose fed *E. coli* cultures the isoleucine/leucine concentration is 300 μM and that ATP is 9.6 mM¹¹. The relative equivalence of these values is in good agreement with calculations from Fersht, whereby an enzyme will

have a K_M close to the free substrate concentration for optimal activity¹². This fits the picture with other class I aaRSs and reflects that *in vivo*, LeuRS may bind Leu independently of ATP. Crucially as stressed in our study the binding of one substrate does not exclude the association of the other, therefore formation of the initial bi-substrate Michaelis complex is not limited by random binding events of either substrate inside the cell.

The confusion came from the different units for the K_M values for Leu and ATP in Table 1. Although not ideal, to our opinion using the same units will make Table 1 more difficult to read.

(2) Superposition with the Leu-bound structure shows that the carboxylate oxygen of Leu is located in the required position to allow nucleophilic attack on the α -phosphate of ATP at a (O-P) distance of 2.9 Å. However, carboxylate ion is usually delocalized and nucleophilicity is reduced compared to the localized form. Therefore, for efficient nucleophilic reaction, proper residue(s) on LeuRS could contribute the localization of the carboxylate ion required. It would be even better if the authors could comment on this point from the structure obtained.

Response: A previous report using a combination of electrostatic potential (ESP) analysis and QM calculations of class I and class II aaRS have argued that the relative positioning of the phosphate backbone of ATP to the carboxylate of the substrate dictates which oxygen atom attacks the alpha phosphorous group¹³. In the same study the authors also showed that the active site residues affect the charge difference between the attacking oxygen and the phosphate to enhance the nucleophilic reaction. Our model is in excellent agreement with these computational studies in that the *syn*-oxygen atom of the Leu carboxylate group is closest to the phosphate as predicted by the computational study. The reviewer raises an important question that can only be delved into deeper with equivalent theoretical QM analyses using our high-resolution structures of NgLeuRS that properly factor in the dramatic domain rearrangements we observed. We believe this is out of remit of the current study but have incorporated a reference to the above paper in our text to highlight these important discoveries:

In line 176 to 181 “Superposition of the Leu-bound structure onto the pre-transition state model shows that the carboxylate oxygen atom which is in the *syn* conformation with the α -amino group of the Leu substrate is located in the required position to allow nucleophilic attack on the α -phosphate of ATP at a (O-P) distance of 2.9 Å (Extended Data Fig. 1d). This positioning in good agreement with theoretical computational models that show the *syn* oxygen atom is the preferred attacking atom in class I aaRSs¹³”

(3) The title of Fig. 4 legend is misleading because Fig. 4(a) did not show the results of L550A/G and E169G mutants. In addition, the labels K_M (Leu) and k_{cat} in the upper panel seem to be hard to understand. The authors should reconsider the points.

Response: This is a valid point raised by the reviewer we have changed the figure legend to:

The legend of figure 4 was corrected as “The effects of L550A and E169G mutants on NgLeuRS catalytic efficiency.”

Regard K_M and k_{cat} in the figure 4a, it was revised as follows:

(4) In Table 1, LeuRS-L550G showed only a 2-fold decrease in the k_{cat}/K_M for Leu activation relative to the WT (first step). However, in the second step (leucine transfer step to tRNA), k_{cat}/K_M value showed about 30-fold drop. In contrast to LeuRS-L550G, the kinetic behaviors in the first and second steps seems to be qualitatively opposite in the case of LeuRS-E169G. It would be better if there was a more sufficient explanation for these differences from the standpoint of their experiments.

Response: The kinetic data show that L550G mutation influences the amino acid activation step (measured by steady-state kinetics and expressed in k_{cat}/K_M constant) significantly less than the isolated transfer step (measured by pre-steady-state kinetics and expressed in k_{trans} constant). As the reviewer correctly recognizes the effect on these steps is opposite for the E169G mutant. In the final paragraph of the results, we have commented that these mutations affect the position of the CP1 hairpin differently: the reduced ability of the L550G mutant to reopen the active site following Leu-AMP formation keeps the active site in the closed conformation, thus influencing tRNA entry and therefore the transfer step. The E169G mutant limits the ability of the CP1 hairpin to close upon formation of the Michaelis complex and therefore influences the activation step. As shown in figure 3A (lower panel) we can measure the unchanged transfer step of the E169G mutant, in spite of the hampered activation step, due to pre-incubation of the enzyme with Leu and ATP to make LeuRS:Leu-AMP intermediate prior mixing with tRNA (the second step is made independent of the rate of the first step by this experiment). To help clarify this further we have expanded the final paragraph of the results to clarify this (lines 313-323):

“The inability of the bound L-leucinol and ATP to induce closure of the CP1 hairpin in the E169G mutant, despite the removal of the L550 clashing sidechain, suggests that the substrates alone cannot promote the closed active site conformation. The interactions mediated by E169 with Y554 and W223 (Fig. 3D) are thus additionally required to stabilize the closed CP1 hairpin position in the pre-transition state to promote the formation of Leu-AMP. In line with this model independently tested amino acid activation demonstrated that substitution of E169 with a glycine results in up to 5-fold increase in K_M (Leu) and 4-fold decrease in k_{cat} yielding a 24-fold less efficient enzyme in Leu activation relative to the WT protein (Table 1 and Extended Data Fig. 3). In contrast to the activation step, the leucyl transfer step is not impaired by the E169G substitution (Table 1). Thus, opposite to the L550G mutant which captures the “closed conformation” and displays a defect in the aminoacyl transfer step, mutation at the CP1 hairpin which stabilizes the “open conformation” affects only the activation step.”

(5) Although the authors concluded that their structural characterization of *N. gonorrhoeae* LeuRS provides a high-resolution structural framework providing the first insights into the amino acid activation mechanism of class Ia aaRSs, historically Alan Fersht has performed extensive works on TyrRS (also class I) in the structural standpoints. The authors should cite more his works other than reference 25 and discuss the relationship with their present results.

Response: The reviewer correctly points out that the both tyrosyl-tRNA-synthetase (TyrRS), and also tryptophanyl-tRNA synthetase (TrpRS), have been well characterized structurally and biochemically. Alan Fersht has indeed made important contributions to the aaRS field in terms of our understanding of the formation of the aminoacyl-adenylate intermediate. However, TyrRS and TrpRS, are both exclusive members of the class Ic subgroup of aaRSs. They differ from LeuRS, which as we state is a class Ia representative, in that they are both dimeric. While the catalytic sites share considerable structural homology, including conservation of the class I HIGH and KMSKS motifs in the catalytic domain both TyrRS and TrpRS also contain an extra GXDQ motif which is additionally associated with substrate binding²⁷⁻²⁹. TyrRS is also known to demonstrate asymmetric (half-of-site) catalysis where only one protomer is active at a time³⁰⁻³². In contrast, class Ia aaRSs comprising LeuRS, IleRS, ValRS, MetRS and CysRS, are monomeric and architecturally distinct from the more historically studied members and therefore the structures we have determined are unique for the subclass Ia and not fully appropriate for class Ic TyrRS and TrpRS. We have though adjusted the introduction to hopefully clarify these distinctions and within it we have added additional appropriate references to Alan Fersht and his peers.

<Minor points>

Line 69, Line 73: Leu- tRNA -> Leu-tRNA (No space between - and t.)

Response: Revised.

Reference 24: tRNA^{Leu} -> tRNA^(Leu)

Reference 55: tRNA^{Ala} -> tRNA^(Ala)

Response: Revised.

Reference 7, 15, 21, 22, 29, 34, 36: Please write using lowercase alphabets, except at the beginning of the title of the papers.

Response: Reference 7, 15, 21, 22, 29, 34, 36 have all been corrected.

Response to Referees References:

1. Lux, M. C., Standke, L. C. & Tan, D. S. Targeting adenylate-forming enzymes with designed sulfonyladenine inhibitors. *J Antibiot (Tokyo)* **72**, 325–349 (2019).
2. Vonrhein, C. *et al.* Data processing and analysis with the autoPROC toolbox. *Acta Crystallogr. D Biol. Crystallogr.* **67**, 293–302 (2011).
3. McCoy, A. J. *et al.* Phaser crystallographic software. *J Appl Crystallogr* **40**, 658–674 (2007).
4. Nautiyal, M. *et al.* Comparative analysis of pyrimidine substituted aminoacyl-sulfamoyl nucleosides as potential inhibitors targeting class I aminoacyl-tRNA synthetases. *Eur J Med Chem* **173**, 154–166 (2019).
5. Emsley, P. & Cowtan, K. Coot: model-building tools for molecular graphics. *Acta Cryst D* **60**, 2126–2132 (2004).
6. Adams, P. D. *et al.* PHENIX: a comprehensive Python-based system for macromolecular structure solution. *Acta Cryst D* **66**, 213–221 (2010).
7. Leatherbarrow, R. J., Fersht, A. R. & Winter, G. Transition-state stabilization in the mechanism of tyrosyl-tRNA synthetase revealed by protein engineering. *Proc. Natl. Acad. Sci. U.S.A.* **82**, 7840–7844 (1985).
8. Fersht, A. R. Dissection of the structure and activity of the tyrosyl-tRNA synthetase by site-directed mutagenesis. *Biochemistry* **26**, 8031–8037 (1987).
9. First, E. A. & Fersht, A. R. Analysis of the role of the KMSKS loop in the catalytic mechanism of the tyrosyl-tRNA synthetase using multimutant cycles. *Biochemistry* **34**, 5030–5043 (1995).
10. Nautiyal, M. *et al.* Comparative analysis of pyrimidine substituted aminoacyl-sulfamoyl nucleosides as potential inhibitors targeting class I aminoacyl-tRNA synthetases. *European Journal of Medicinal Chemistry* **173**, 154–166 (2019).
11. Bennett, B. D. *et al.* Absolute metabolite concentrations and implied enzyme active site occupancy in *Escherichia coli*. *Nat Chem Biol* **5**, 593–599 (2009).
12. Baldwin, R. L. Structure and mechanism in protein science. A guide to enzyme catalysis and protein folding, by A. Fersht. 1999. New York: Freeman. 631 pp. \$67.95 (hardcover). *Protein Science* **9**, 207–207 (2008).

13. Banik, S. D. & Nandi, N. Mechanism of the activation step of the aminoacylation reaction: a significant difference between class I and class II synthetases. *Journal of Biomolecular Structure and Dynamics* **30**, 701–715 (2012).

REVIEWERS' COMMENTS:

Reviewer #2 (Remarks to the Author):

The authors have satisfactorily responded to all my questions and made the necessary changes to the manuscript.